# Chemically programmed STING-activating nano-liposomal vesicles improve anticancer immunity

Xiaona Chen[1], Fanchao Meng[1], Yiting Xu[1], Tongyu Li[1], Xiaolong Chen[1] & Hangxiang Wang [1,2] ✉

The often immune-suppressive tumor microenvironment (TME) may hinder immune evasion and response to checkpoint blockade therapies. Pharmacological activation of the STING pathway does create an immunologically hot TME, however, systemic delivery might lead to undesired off-target inflammatory responses. Here, we generate a small panel of esterase-activatable pro-drugs based on the structure of the non-nucleotide STING agonist MSA-2 that are subsequently stably incorporated into a liposomal vesicle for intravenous administration. The pharmacokinetic properties and immune stimulatory capacity of pro-drugs delivered *via* liposomes (SAProsomes) are enhanced compared to the free drug form. By performing efficacy screening among the SAProsomes incorporating different pro-drugs in syngeneic mouse tumor models, we find that superior therapeutic performance relies on improved delivery to the desired tumor and lymphoid compartments. The best candidate, SAProsome-3, highly stimulates secretion of inflammatory cytokines and creates a tumoricidal immune landscape. Notably, upon application to breast cancer or melanoma mouse models, SAProsome-3 elicits durable remission of established tumors and postsurgical tumor-free survival while decreasing metastatic burden without significant systemic toxicity. In summary, our work establishes the proof of principle for a better targeted and more efficient and safe STING agonist therapy.

Cancer immunotherapy using immune checkpoint blockade (ICB) inhibitors (e.g., anti-PD-1/PD-L1 and anti-CTLA-4 antibodies) has achieved remarkable clinical success, yielding durable and long-term therapeutic responses in multiple cancer types[1]. However, only a small subset of patients benefit from this treatment[2]. Many inaccessible tumors are immunologically cold, characterized by the abundant infiltration of immune suppressors while being devoid of tumor-infiltrating lymphocytes, thus enabling their escape from immune surveillance[3,4]. Consequently, most cancers exhibit overwhelming de novo refractoriness to the ICB antibodies approved by the Food and Drug Administration[5–7]. Therefore, effective immunotherapeutic approaches beyond those directly targeting the adaptive immune response are expected to benefit large populations of patients with cancer, and hence, are desperately needed[8–11].

Stimulator of interferon genes (STING) is an intracellular signaling receptor that regulates the innate immune pathway and is critical for the initiation of antitumor immunity[12]. The pharmacological activation of STING via endogenous cyclic dinucleotides (CDN), such as

[1]The First Affiliated Hospital, NHC Key Laboratory of Combined Multi-Organ Transplantation, Collaborative Innovation Center for Diagnosis and Treatment of Infectious Diseases, State Key Laboratory for Diagnosis and Treatment of Infectious Diseases, Zhejiang University School of Medicine, 310003 Hangzhou, Zhejiang Province, P. R. China. [2]Jinan Microecological Biomedicine Shandong Laboratory, 250117 Jinan, Shandong Province, P. R. China. ✉e-mail: wanghx@zju.edu.cn

2′,3′−cyclic guanosine monophosphate–adenosine monophosphate (cGAMP), which prompts the induction of type-I interferons (IFN) and other proinflammatory cytokines. This further stimulates dendritic cell (DC) activation and the cross-presentation of tumor antigens, reversing tumor immune desertification[13,14]. Substantial efforts have been focused on the development of CDN derivatives that mimic endogenous cGAMP[15–17]. However, the efficacy of these agonists is considerably compromised because of metabolic instability and rapid clearance from the body following systemic administration. The therapeutic delivery of CDN-based STING agonists into the cytosol of target cells is difficult because of the high hydrophilicity and negative charge of the molecule[18,19]. Furthermore, intravenously injected small-molecule STING agonists can engender uncontrolled systemic dissemination and widespread inflammatory responses[20]. Thus, current clinical trials involving CDN-based STING agonists are focused on direct intratumoral injection, which limits their clinical implementation to patients with accessible solid tumors[21,22]. However, the stimulation of antitumor immunity via an intravenous systemic regimen imparts a notable advantage in eliminating surgically unresectable, and particularly, metastasized cancer lesions[23]. The abovementioned challenges have provoked research into STING-activating systems for the conduction and rapid expansion of their immunotherapy trials to combat cancer. MSA-2 is a recently developed non-nucleotide STING-activating agonist. This agent has been administered orally in animal studies, although low oral bioavailability and inadequate cytosolic entry may limit its antitumor efficacy[24]. We hypothesized that the carboxyl moiety (10-OH) on the MSA-2 molecule may account for its poor compatibility with drug carriers and that this could be improved via rational chemical derivatization.

Here we develop STING-activating liposomal vesicles that entrap engineered MSA-2 pro-drugs for leveraging innate antitumor immunity (Fig. 1a–c). The drug activation kinetics of these morpholine (MP)-type pro-drug entities are tunable via varying hydrocarbon numbers, thus allowing the selection of candidates that can rapidly release active drugs for effective STING activation. We disclose a remarkably robust structure–activity relationship for engineered STING agonist pro-drug liposomes (SAProsomes), enabling their optimization in terms of adaptive formulation and in vivo antitumor efficacy. Notably, on administration to triple-negative breast cancer 4T1 and melanoma cancer B16F10 models, which are largely refractory to ICB therapy, SAProsome primes antitumor immunity, durable remission of established tumors, postsurgical tumor-free survival, and inhibition of metastatic burden. Hence, this study highlights the potential of STING pro-drugs optimized via liposomal delivery as promising treatments for improving antitumor immunity.

## Results

### Rational design of STING-activating pro-drugs for liposomal formulation

A recently discovered non-nucleotide STING agonist, MSA-2, was selected to test the validity of our drug design. To facilitate the liposomal formulation of MSA-2, four synthetic MSA-2 derivatives (compounds **1**–**4**) were initially constructed via ester bonds using MP-typed alkanols of varying lengths as modifiers (Fig. 1a; for synthesis details, see Supplementary Information). Their chemical structures were unambiguously confirmed using proton nuclear magnetic resonance and high-resolution mass spectrometry (Supplementary Figs. S1–S10). We attempted to reduce the polarity of the carboxyl group on MSA-2 via chemical derivatization using MP to enable the efficient assembly of the pro-drugs into liposomal nanovesicles. The ester bond is susceptible to esterase-catalyzed hydrolysis in tumors, thereby enabling the in situ release of chemically unmodified agonists for STING activation[25–28]. We then investigated the esterase-triggered activation of MSA-2 by the pro-drugs[29]. MSA-2 pro-drugs exhibited tunable and linker length-dependent hydrolytic kinetics in the presence of porcine

liver esterase (PLE) (Fig. 1d). Although all the pro-drugs were resistant to hydrolysis in phosphate-buffered saline (PBS) buffer at 37 °C in the absence of PLE, most pro-drugs (~90%) were converted into chemically unmodified STING agonists within 30 min in the presence of esterase in vitro. The pseudo-first−order rate constants (± s.d.) for STING agonist release from these pro-drugs were $31.9 \pm 0.3 \times 10^{-3}$ min$^{-1}$ (pro-drug **1**), $23.9 \pm 3.5 \times 10^{-3}$ min$^{-1}$ (pro-drug **2**), $64.1 \pm 0.2 \times 10^{-3}$ min$^{-1}$ (pro-drug **3**), and $199.1 \pm 1.2 \times 10^{-3}$ min$^{-1}$ (pro-drug **4**) (Fig. 1e, f).

Next, we tested the ability of these MSA-2 pro-drugs to assemble into liposomal nanovesicles (SAProsomes). The parent MSA-2 molecule was immiscible with liposomal components, producing precipitates exclusively in aqueous solutions. Conversely, following the nanoprecipitation protocol, MSA-2 pro-drugs **1**–**4** were highly compatible with liposomal components, with a loading efficiency of >90% for the different pro-drugs (Table S1). Typical nanosized liposomal structures were formed for these pro-drug-loaded nanoparticles, as evidenced by transmission electron and cryo-electron (Fig. 1g) microscopy observations[30]. Furthermore, these SAProsomes exhibited a suitable z-average diameter of ~120 nm, as measured by dynamic light scattering, and a narrow size distribution, as reflected by a low polydispersity index. All SAProsomes, except SAProsome-4, remained sufficiently stable in buffered solutions with serum (20%, w/v) (Fig. 1h, i).

### SAProsomes enhanced the cytosolic delivery of STING agonists and potentiated STING activation

Subsequently, whether SAProsomes could facilitate the cellular uptake and preserve the immunostimulatory activity of STING agonists was investigated. The intrinsic fluorescence of MSA-2 was evaluated to quantify the intracellular drug concentration, as the fluorescence spectra remained unaffected irrespective of chemical derivatization (Supplementary Fig. S11). Exposing cells to SAProsomes engendered substantially higher MSA-2−derived fluorescence signals than free MSA-2 exposure (Fig. 2a, b), suggesting the cellular uptake of MSA-2 was accelerated via liposomal delivery. We further investigated whether increased cytosolic delivery contributed to higher inflammatory signaling. The in vitro stimulation of bone marrow-derived dendritic cells (BMDC) using SAProsome increased the mRNA expression of downstream STING-regulated cytokines, such as IFN-β, tumor necrosis factor alpha (TNFα), and chemokine (C-X-C motif) ligand 10 (CXCL10), whereas free MSA-2 was less effective at increasing mRNA expressions of downstream cytokines (Fig. 2c). CXCL10 has been implicated as a chemoattractant for promoting the intratumoral infiltration of T cells, whereas IFN-β and TNFα are crucial immunomodulators for enhancing antitumor responses[31–33]. In addition, dose-dependent secretion of the critical cytokine, IFN-β, after treatment with SAProsomes was observed in BMDCs (Fig. 2d), which allowed to identify the superior activity of SAProsome-3 and 4 with relatively low half maximal effective concentration (EC$_{50}$). We further validated their potential of STING activation in human monocytic cell line THP1. Consistent with the in vitro results, active MSA-2 was spontaneously released from SAProsome-3 following cellular uptake (Supplementary Fig. S12). In addition, SAProsome-3 treatment not only upregulated mRNA expressions of IFN-β, TNFα and CXCL10, but also triggered STING downstream signaling cascades, inducing phosphorylation of STING protein, TANK-binding kinase 1 (TBK1), and IFN regulatory factor 3 (IRF-3) in THP1 cells (Supplementary Fig. S13).

Given the superior activity of SAProsomes in stimulating the STING pathway and IFN-β secretion, we further investigated whether this stimulation induced DC maturation in vitro. DC maturation analysis revealed that SAProsomes significantly upregulated the expression of major histocompatibility complex II (MHC II) and costimulatory markers of CD80/CD86 in BMDCs (Fig. 2e, f). The overall induction of DC maturation via all the SAProsomes displayed a comparable response, although SAProsome-4 exhibited superior cellular uptake efficiency and IFN-β secretion. DC maturation is accompanied by an

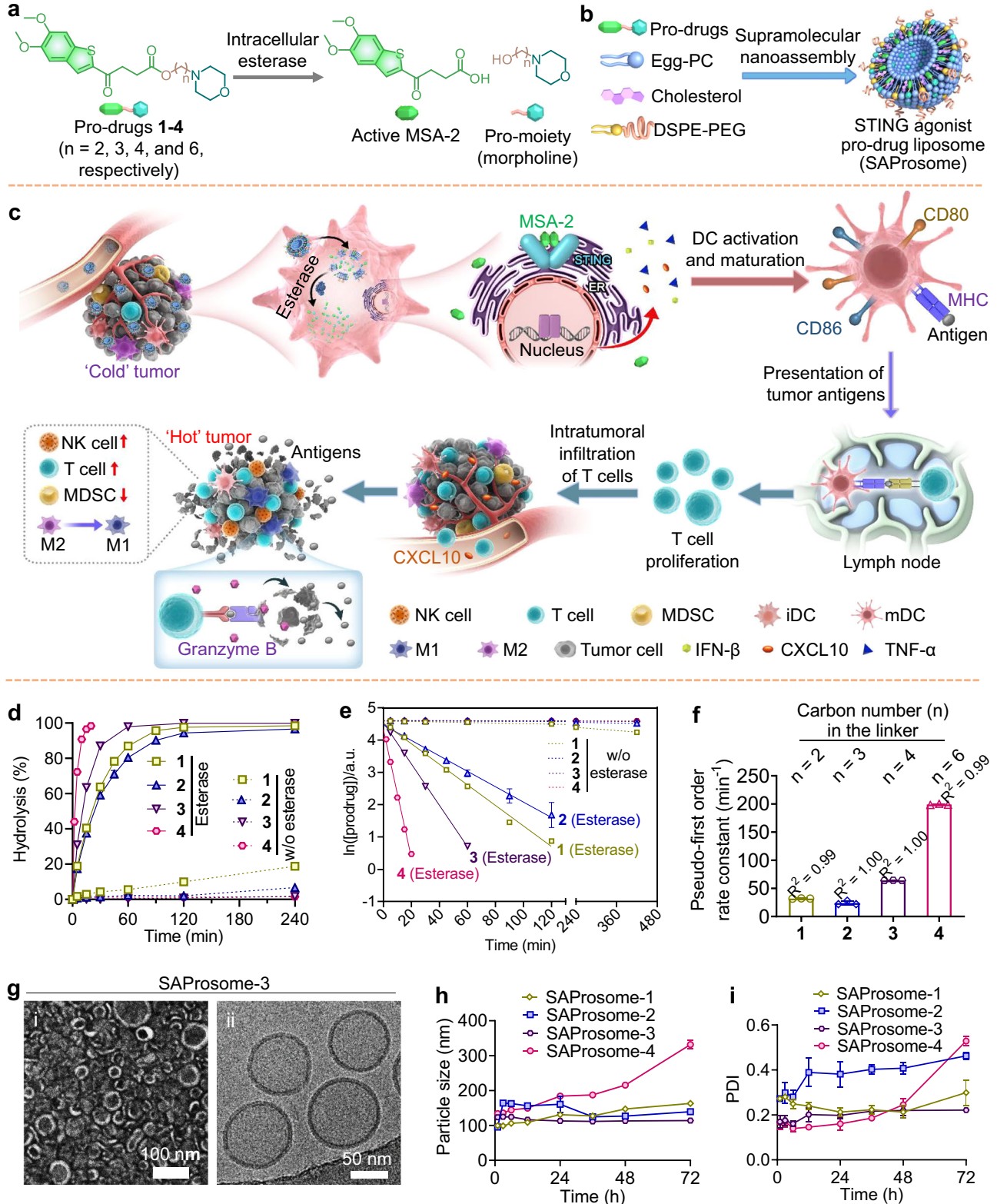

**Fig. 1 | Stimulator of interferon gene (STING)-activating liposomal vesicles (SAProsomes) for potentiating anticancer immunity. a, b** Rational design of MSA-2−derived STING pro-drugs **1**−**4** via hydrolytic ester bond for optimization of in vivo liposomal delivery. The pro-drugs were designed to be hydrolyzed by esterase to spontaneously release active MSA-2 in the target cells. Unlike free MSA-2, these pro-drugs can be stably assembled into lipid constituents to form liposomal vesicles. **c** Schematic illustration of esterase-responsive activation of MSA-2 as a STING-activating molecule followed by antitumor immune stimulation in the tumor-bearing mice. **d** Drug activation kinetics from pro-drugs **1**−**4** in the presence or absence of porcine liver esterase (PLE, 50 unit/mL) in phosphate-buffered saline (PBS) at 37 °C. Data are presented as ± s.d. of the mean, $n = 3$. **e, f** Extrapolation of pseudo-first-order rate constants from the curves in (**e**) when the pro-drugs were incubated in PLE-containing PBS. Data are presented as ± s.d. of the mean, $n = 3$. **g** Representative transmission electron microscopy (TEM) and Cryo-TEM images of SAProsome-3. Each experiment was independently repeated in triplicates, yielding similar results. **h, i** Stability of SAProsomes. Particle sizes (**h**) and polydisperse index (PDI) (**i**) were monitored via dynamic light scattering analysis over 72 h. Data are presented as ± s.d. of the mean, $n = 3$. Source data are provided as a Source data file.

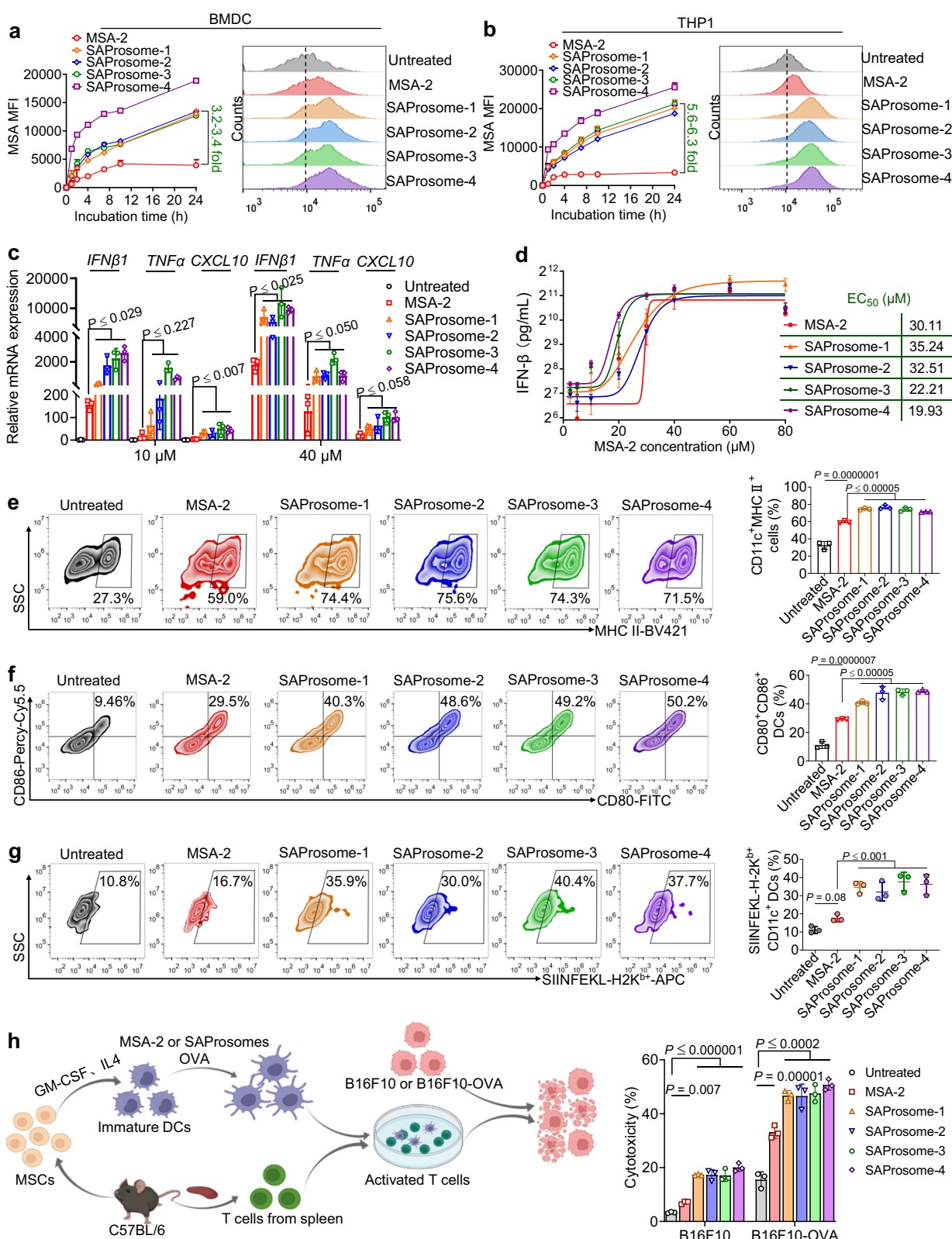

enhanced antigen presenting capacity[34]. Therefore, we evaluated the ability of SAProsomes to promote the engulfed antigen by cross-presenting it to BMDC surfaces. BMDCs exposed to SAProsomes significantly improved SIINFEKL-H-2Kb presentation by 2.9–3.4- and 1.8–2.1-fold compared with the presentation in cells treated with saline and free MSA-2, respectively (Fig. 2g). Crucially, pretreatment of BMDCs with SAProsomes and a tumor-specific antigen (e.g., OVA)

significantly enhanced the cytotoxicity of splenic T cells against tumor cells (B16F10-OVA), as compared to untreated cells (Fig. 2h). In contrast, splenic T cells had limited cytotoxicity against B16F10 tumor cells. These experimental results indicate that intracellular esterase can spontaneously trigger the liberation of active MSA-2, which eventually facilitate the activation of STING pathway and lead to augmented secretion of cytokines.

**Fig. 2 | STING-activating liposomal vesicles (SAProsomes) increase the intracellular uptake of MSA-2 and stimulate dendritic cell (DC) maturation in vitro. a, b** Profiles of intracellular MSA-2 fluorescence intensity quantified using flow cytometry (left panel). SAProsomes improve the cytosolic delivery of MSA-2 in bone marrow-derived DCs (BMDCs) (**a**) or THP-1 cells (**b**). Cells were incubated with free MSA-2 or different SAProsomes (40 μM MSA-2 equivalence). Representative flow cytometric histograms showing the uptake of fluorescent MSA-2 or pro-drugs following a 24-h incubation (right panel). Data are presented as ± s.d. of the mean, $n = 3$ biologically independent cells. **c** STING-dependent downstream expression of inflammatory cytokines in BMDCs treated with MSA-2 or SAProsomes as assessed via qRT-PCR ($n = 3$). **d** Dose−response curves of IFN-β secretion upon various treatments. The concentration of IFN-β in BMDC culture medium was determined via enzyme-linked immunosorbent assay (ELISA). Data are presented as ± s.d. of the mean, $n = 3$ biologically independent samples. **e, f** Following treatment with SAProsomes, BMDCs exhibit significantly increased activation markers, including major histocompatibility complex II (MHC II) (**e**) and CD80/CD86 (**f**) ($n = 3$). **g** Augmented antigen cross-presentation of BMDCs induced by SAProsomes. Percentages of ovalbumin (OVA, SIINFEKL)-presenting cells were determined using flow cytometry ($n = 3$). **h** Cotreatment with the OVA peptide and SAProsomes increased the B16F10-OVA cell lysis via cytotoxic T lymphocytes in vitro. Experimental protocol is shown on left-hand side, which was created using BioRender.com. Cell cytotoxicity was determined based on the lactate dehydrogenase (LDH) concentration in medium supernatant, as measured by the LDH assay ($n = 3$ biologically independent samples). The cell number ratios (T cell/B16F10 or B16F10-OVA cell/BMDC) were fixed at 10:1:5. In (**c**, **e**−**h**), data are presented as the mean ± s.d. and statistically analyzed using one-way analysis of variance. Source data are provided as a Source data file.

## Systemic SAProsome treatment outperformed free MSA-2 and eliminated established tumors

Liposomal vesicles of the MSA-2 pro-drug enabled systemic administration via intravenous injection. A preclinical mouse model bearing subcutaneous colorectal MC38 tumor was first included to test the antitumor efficacy of these SAProsomes in comparison with free MSA-2 that is orally available (Fig. 3a). Significant tumor shrinkage was observed after three intravenous injections of SAProsomes, whereas dosing of free MSA-2 at 35 mg/kg failed to impede tumor growth. This result was further supported by the tumor growth kinetics monitored each other days (Fig. 3b−f). As a comparison, empty liposomes without pro-drug payloads did not show any antitumor benefits in this model (Supplementary Fig. S14). SAProsomes appeared to be safe and well tolerated in animals, as evidenced by the stable growth of body weights of the mice (Fig. 3c). SAProsome treatment engendered complete tumor regression (CR), and the tumor inhibition rates were highly dependent on the encapsulated MSA-2 pro-drugs. Notably, SAProsome-3 outperformed the other SAProsomes with 100% CR rate and all mice ($n = 10$) remained tumor-free at the study endpoint (Fig. 3d). Hematoxylin and eosin (H&E) staining of tumor sections revealed higher necrotic areas in the tumors of mice treated with SAProsome-3, which was further verified by TUNEL immunohistochemical analysis (Fig. 3g and Supplementary Figs. S15 and 16). However, remarkably, tumors in mice treated with free MSA-2 harbored few CD8+ T cells, whereas the tumors of SAProsome-3−treated mice were abundantly infiltrated with CD8+ T cells and accompanied by the profuse diffusion of granzyme B, a defining feature of T cell function (Fig. 3g and Supplementary Fig. S17). Moreover, SAProsome treatment effectively increased the survival time compared with that observed with free MSA-2 administration (median survival time [MST]: MSA-2 = 15 days; SAProsomes-1, 2, and 4 = 93−99 days) (Fig. 3h). Consistent with the tumor growth curves, SAProsome-3 substantially extended the MST of tumor-bearing mice over 150 days.

To further evaluate the immunological memory effects, SAProsome-3−treated mice surviving on day 150 following the first MC38 tumor cell inoculation were rechallenged with MC38 cells[35]. Compared with the rapid tumor growth in the treatment-naïve mice, all SAProsome-3−treated mice lacked secondary tumor reoccurrence during the 60-day observation period, thus indicating the establishment of a durable tumor-specific immunological memory (Fig. 3i).

## Immunoregulatory efficacy of SAProsome-3

We next analyzed the difference in immunoregulatory effects in the tumor tissues of mice treated with SAProsome-3 and free MSA-2 via flow cytometry. SAProsome-3 administration significantly expanded CD3+CD8+ cytotoxic T cell subsets in tumors compared with free MSA-2 treatment (Fig. 4a and Supplementary Fig. S18). The CD8+/CD4+ T cell ratio, a critical prognostic indicator for immunotherapy outcomes,

significantly increased following SAProsome-3 treatment versus that in the untreated group (2.82 versus 0.26, $P = 0.0000004$)[36]. Multiple effector CD4+ and CD8+ T cell populations were highly enriched in the tumors of mice immunized with SAProsome-3 (Fig. 4b). In particular, there was a significant increase in activated IFN-γ+CD8+ T cells in SAProsome-3−treated mice compared with that observed in the untreated or MSA-2 groups (37.2% versus 15.7% or 18.0%, $P = 0.000002$ or $P = 0.000009$, respectively). Moreover, the surface expression of CD206—a canonical marker of M2-polarized macrophages—dramatically decreased in the macrophages of SAProsome-3−treated mouse tumors (Fig. 4c and Supplementary Fig. S19), suggesting the repolarization or recruitment of macrophages during reduced immunosuppressive activity[37,38]. The influx of tumor-infiltrating natural killer (NK) cells was significantly increased during SAProsome-3 treatment than that during free MSA-2 treatment (Fig. 4d). Furthermore, increased DC expression of costimulatory CD80/CD86 was observed in the TME, consistent with the substantial increase in the tumor CD8+ T cells in SAProsome-3−treated mice (Fig. 4e and Supplementary Fig. S20). Myeloid-derived suppressor cells (MDSC) are known to exhibit immunosuppressive activity and are important in tumor immune evasion. Here, SAProsome-3 treatment substantially reduced the infiltration of intratumoral MDSCs compared with that in other treatments (Fig. 4f, g).

To further assess the ability of SAProsome-3 to activate antigen-presenting cells in vivo, we isolated tumor-draining lymph nodes (TDLN) from mice for flow cytometric analysis. The proportion of matured DCs, as indicated by CD80+/CD86+ DCs, in the TDLN of SAProsome-3−treated mice was 3.0- or 2.1-fold greater than that of mice treated with saline or free MSA-2, respectively (Fig. 4h). Intravenously injected SAProsome-3 significantly elevated the number of CD8+ T cells and promoted M2-to-M1 repolarization of macrophages in TDLNs (Fig. 4i, j). Furthermore, significantly elevated levels of CD8+ T cells and maturation of robust DCs as well as promotion of M2-to-M1 repolarization in the spleen were observed (Fig. 4k−m). Collectively, these results provide compelling evidence that SAProsome-3 triggers robust immune activation, thus contributing to sustained tumor regression in the preclinical MC38 model used here.

## Efficacy of SAProsome-3 in treating postsurgical cancer metastases

One of the most challenging problems encountered in oncology is the recurrence of cancer due to metastases, which occur in numerous patients despite the successful removal of the primary tumor. Therefore, we assessed the therapeutic efficacy of SAProsome-3 in this clinically important setting following the surgical resection of primary and early metastatic tumors. To this end, an orthotopic triple-negative breast cancer model with spontaneous metastasis was established by inoculating luciferase-expressing 4T1 (4T1-luc) cells into the mammary fat pads of

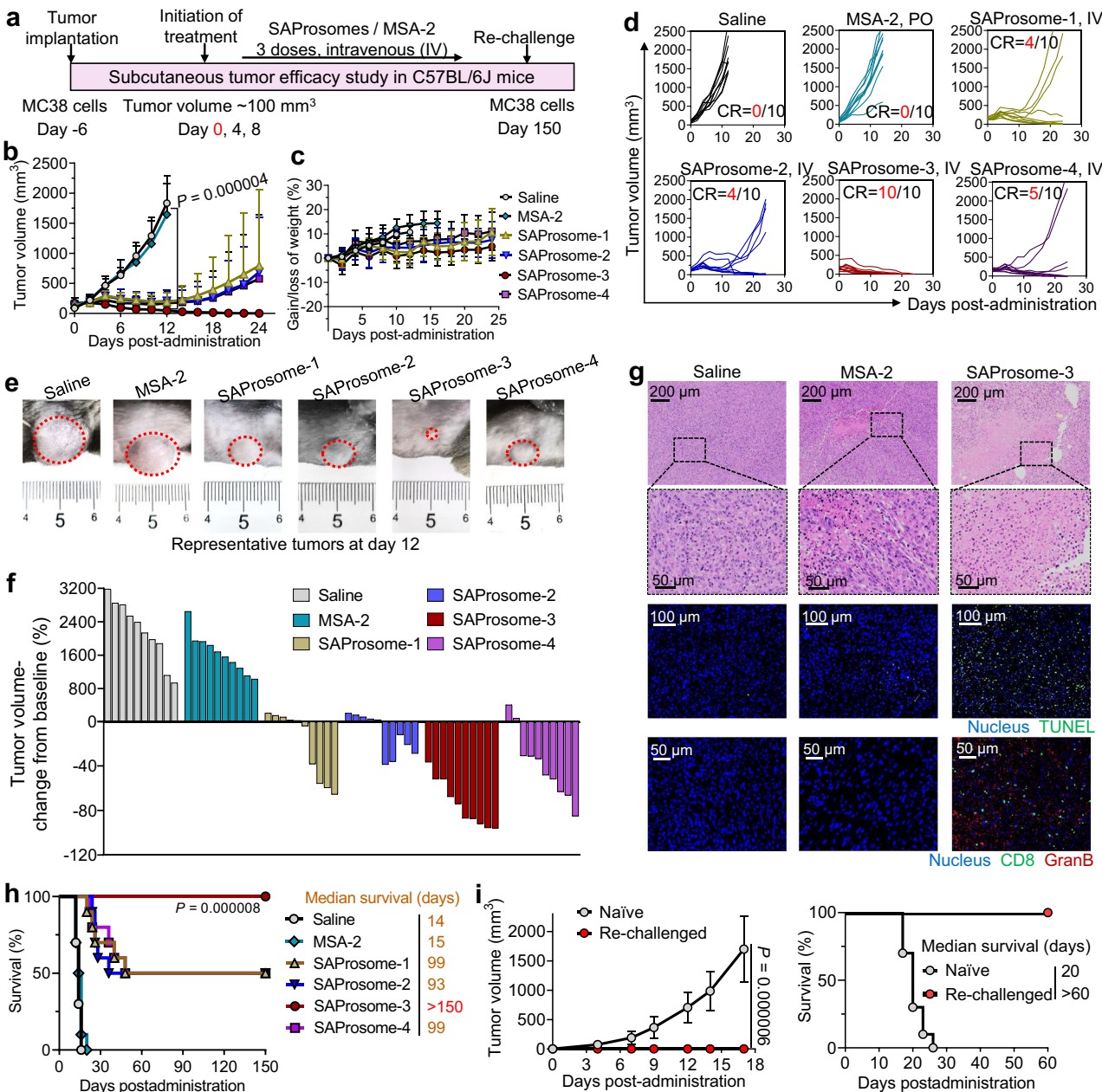

**Fig. 3 | Intravenously administered SAProsomes induced tumor regression in established colorectal MC38 tumors. a** Schematic of the experimental design involving a syngeneic mouse tumor model. Mice with -100 mm³ subcutaneous tumors were intravenously administered with SAProsomes thrice at an MSA-2–equivalent dose of 35 mg/kg. Free MSA-2 administered via oral gavage or saline was included for comparison. Tumor-growth kinetics (**b**) and body weight changes (**c**) in the different groups (*n* = 10 mice/group). Data are presented as the mean ± s.d. and statistically analyzed using two-tailed Student's *t* test. **d** Tumor-growth curves for individual mice (CR, complete responder). **e** Representative photographs of MC38 tumors on day 12. **f** Waterfall plot showing the response of MC38 tumors to different treatments after 12 days (values represent volume-change compared with the baseline before treatment). **g** Representative hematoxylin and eosin staining, terminal deoxynucleotidyl transferase-mediated dUTP nick end labeling (TUNEL) analysis, and immunofluorescence assay of cytotoxic CD8⁺ T cell in excised tumors on day 10 after treatment. Triplicates were performed independently with similar results. **h** Kaplan–Meier survival analysis (*n* = 10 mice/group). Statistical significance was determined using log-rank (Mantel–Cox) tests. **i** Tumor-growth curves and Kaplan–Meier survival analysis after reinoculation of MC38 cells in mice, showing no reoccurrence of the secondary tumor in SAProsome-3–treated mice (*n* = 10/group). Treatment-naïve mice (*n* = 10/group) were included as control. Data are presented as the mean ± s.d. and statistically analyzed using two-tailed Student's *t* test. Source data are provided as a Source data file.

immunocompetent syngeneic BALB/c mice (Fig. 5a)[39]. SAProsome-3 effectively induced the sustained regression of primary tumors (Fig. 5b) with stable body weight growth (Supplementary Fig. S21) in this model; this result was further supported by negative bioluminescence in the treatment group (Fig. 5c and Supplementary Fig. S22). On day 13, orthotopic tumors were surgically resected and postsurgery in vivo bioluminescence imaging was subsequently conducted to visualize the metastatic burden (Fig. 5d). SAProsome-

3 demonstrated a marked therapeutic efficiency in inhibiting tumor metastases with no detectable 4T1 cell-derived bioluminescence signals compared with those observed in mice treated with MSA-2 only. Importantly, the survival rate in the SAProsome-3 treatment group underwent a significant increase to 100% (*n* = 5) within 120 days (Fig. 5e).

We additionally examined the inhibitory effects of SAProsome-3 on tumor metastatic burden. Organ weight has been correlated with

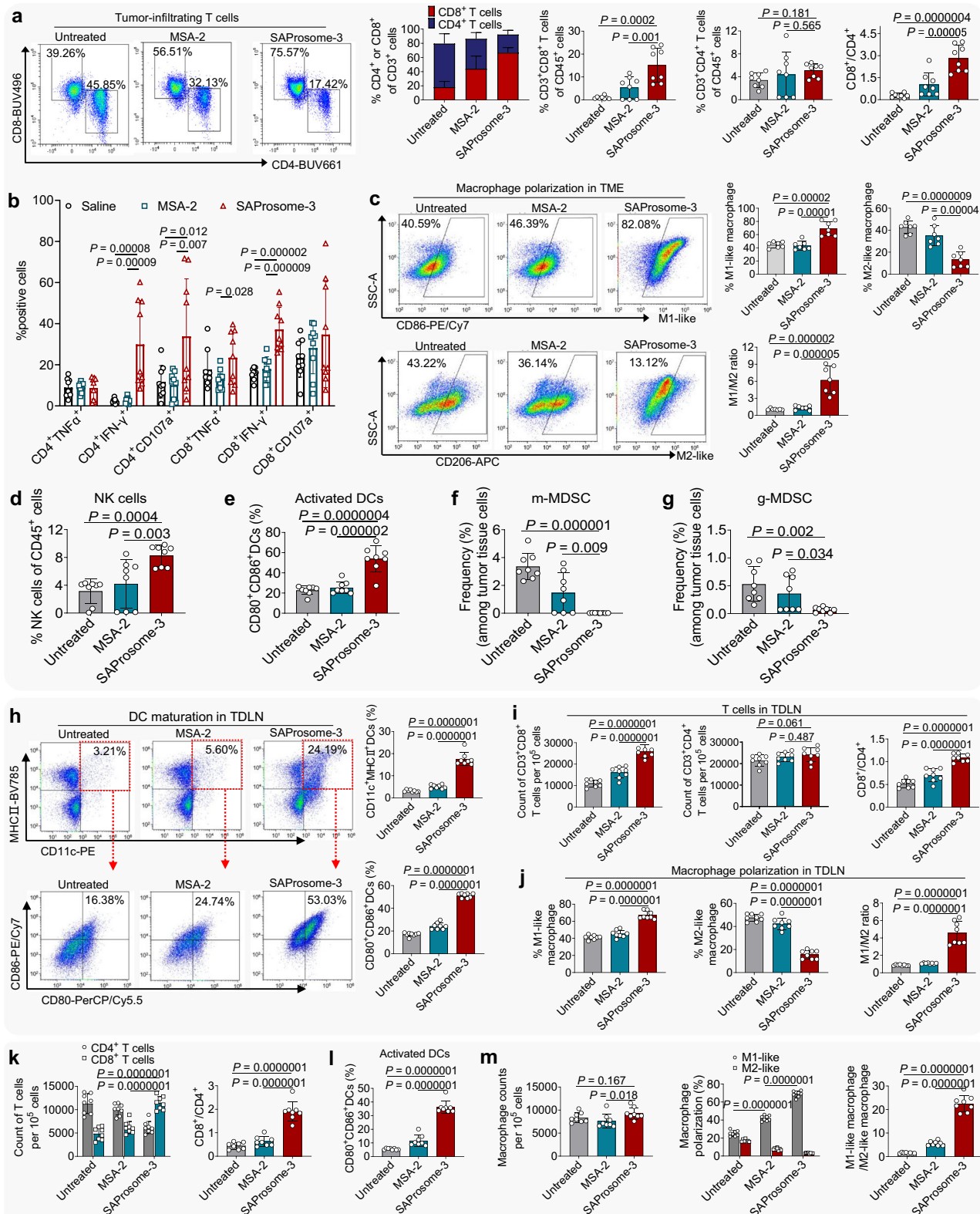

metastasis outgrowth. We observed significantly decreased organ weights, such as those of the liver, lungs, and spleen, and reduced macrometastases in SAProsome-3–treated mice (Fig. 5f–h). Furthermore, SAProsome-3 treatment produced smaller TDLN weights compared with those observed in other treatments (Fig. 5i). The organ sections were histologically analyzed by H&E staining to further examine the metastasis. Encouragingly, the organs in the mice treated with SAProsome-3 displayed normal morphology, and no recurrence or metastatic nodules were observed (Fig. 5j). Hence, we concluded that SAProsome-3 exhibits considerable potential for preventing metastatic relapse, thereby reducing cancer-related mortality and prolonging survival.

**Fig. 4 | SAProsome-3 enhanced cytokine production and reprogrammed the immunosuppressive tumor microenvironment (TME) in the MC38 tumor model. a** Flow cytometric plots of T cells in MC38 tumor tissue analyzed on day 10 after treatment (gating on CD3$^+$ cells, $n = 8$ mice/group for biologically independent samples). **b** Intracellular cytokine staining to evaluate TNFα and IFN-γ production by tumor-infiltrating CD4$^+$ and CD8$^+$ T cells in response to phorbol myristate acetate/ionomycin stimulation and the frequency of CD107a$^+$ T cells in the tumors were analyzed using flow cytometry ($n = 9$ biologically independent samples). **c** Flow cytometric plots and quantitative analysis of M1- and M2-like macrophages within the TME ($n = 7$ biologically independent samples). **d, e** Cell proportions of NK cells in CD45$^+$ cells (**d**) and dendritic cells (DC) expressing CD80/CD86 (**e**) in tumors ($n = 8$/group). **f, g** Frequency of granulocytic and monocytic myeloid-derived suppressor cells (MDSC) (gMDSC and mMDSC, respectively) in tumors ($n = 8$ biologically independent samples). **h** Flow cytometry scatter plots and quantification of CD80/CD86 expression by DCs in the tumor-draining lymph nodes (TDLN) ($n = 8$ biologically independent samples). **i** Frequency of CD3$^+$CD8$^+$ and CD3$^+$CD4$^+$ T cells and the ratio of CD8$^+$ cytotoxic T cells versus CD4$^+$ T cells ($n = 8$ samples/group) in TDLNs after treatment. **j** Macrophage polarization profiles in TDLNs ($n = 8$ biologically independent samples). **k–m** Splenocytes from mice in each treatment group were analyzed on day 10 using flow cytometry to examine the frequency of CD4$^+$ and CD8$^+$ T cells (**k**) activated DCs expressing CD80/CD86 costimulatory molecules (**l**) and M1- and M2-like macrophages (**m**) were also analyzed in the spleen. $n = 8$ biologically independent samples. In **a–m**, data are presented as the mean ± s.d. and statistically analyzed using one-way analysis of variance. Source data are provided as a Source data file.

## Efficacy comparison of SAProsome-3 versus free MSA-2 via different routes

We next sought to compare the therapeutic efficacy of SAProsome-3 versus free MSA-2. The mouse tumor model bearing Lewis lung carcinoma (LLC) was established and the animals were treated with different doses and administration routes of therapeutics. Dose-dependent antitumor activity was observed in each treatment group (Fig. 6a and Supplementary Fig. S23). Interestingly, in this model, we still observed the striking potency of intravenous SAProsome-3, yielding durable tumor regression and complete responses in 50 and 66.7% of mice at 35 and 40 mg/kg of MSA-2–equivalent doses, respectively (Fig. 6a, b). Notably, dosing of SAProsome-3 at 40 mg/kg led to significant tumor volume shrinkage to ~30 mm$^3$. In sharp contrast, free MSA-2 administered either by oral or intravenous route only showed limited efficacy against LLC tumors (Fig. 6a–c and Supplementary Fig. S23). We also confirmed the doses of oral MSA-2 at 160 mg/kg, intravenous MSA-2 at 40 mg/kg, and intravenous SAProsome-3 at 30 mg/kg (MSA-2 equivalence), by which they had the comparable antitumor efficacy in this animal model (Fig. 6a–c). Histological analysis using H&E staining and TUNEL-positive cells labeling revealed pronounced intratumoral apoptosis after SAProsome-3 treatment (Supplementary Fig. S24). Moreover, administration of SAProsome-3 enhanced infiltration of CD8$^+$ T cells in the tumors, which correlated positively with elevated production of granzyme B (Supplementary Fig. S24). Again, SAProsome-3 was proven to be safe for intravenous administration as supported by stable body weights in animals (Fig. 6d).

## Combination therapy of SAProsome-3 with immune checkpoint blockade

The blockade of programmed cell death protein 1 (PD-1) or its ligand (PD-L1) with respective antibodies have shown tremendous promises in the clinic. Moreover, blocking the PD1/PD-L1 axis in combination with other treatment modalities can yield additional therapeutic benefits in cancer treatment. We therefore assessed the efficacy of optimized SAProsome-3 in combination with PD-L1–based ICB in a syngeneic murine tumor model, an immunologically cold and poorly ICB-responsive tumor-B16F10 melanoma in C57BL/6 mice[40]. The treatment was initiated when the volume of subcutaneous tumors reached 100–200 mm$^3$. The administration of monotherapy to these established tumors with either an anti-PD-L1 antibody (αPD-L1) or free MSA-2 only partly retarded their growth (Fig. 6e, f). Remarkably, SAProsome-3 treatment led to robust tumor suppression compared with that observed with free MSA-2 treatment (e.g., tumor-growth inhibition on day 10 of 92.3% and 24.4% with SAProsome-3 and MSA-2 treatments, respectively) (Fig. 6e)[41]. Notably, the combination therapy eradicated large established tumors in mice by day 10 after SAProsome-3 and αPD-L1 administration, while the efficacy lasted across the observation period (Fig. 6e, f). Mice in all the groups maintained normal body weight growth, indicating that the treatments did not induce severe toxicity (Fig. 6g). Moreover, the MST of mice treated with the combination therapy was prolonged to 27 days

compared with that in other groups (e.g., MST saline = 10 days) (Fig. 6h)[42]. These results support the synergy between PD-L1 inhibition and SAProsome-3–mediated STING activation in eliciting robust systemic antitumor immunity.

## Liposomal delivery improves the pharmacokinetic properties and biodistribution of STING agonists

We further explored the mechanism of liposomal delivery to improve the antitumor immunity observed in multiple in vivo models by investigating the pharmacokinetic profiles and drug distribution in tumors and lymph nodes. The pharmacokinetic profiles of SAProsome in Sprague-Dawley rats following single intravenous injection were determined by measuring plasma drug concentrations[43]. Free MSA-2 administered either orally or intravenously was included for comparison. The plasma concentration of total MSA-2 versus time and related pharmacokinetic parameters are shown in Fig. 7a, b, respectively. Remarkably, the liposome-delivered MSA-2 pro-drugs exhibited prolonged systemic circulation with half-lives ranging from 4.61 to 5.95 h, 5.6–7.3-fold higher than those observed after intravenous MSA-2 injection ($t_{1/2} = 0.82$ h). Notably, the area under the concentration versus time curve ($AUC_{0-t}$) revealed that SAProsome-3 demonstrated the highest $AUC_{0-t}$, 10.6-fold greater than that observed with orally administered MSA-2.

To clarify whether the improved pharmacokinetic property of SAProsomes can benefit favorable drug distribution in targeted tissues, the concentration of MSA-2 in both the tumors and lymph nodes was investigated following treatment in multiple tumor-bearing mouse models. Free MSA-2 administered via either intravenous injection or gavage showed comparatively low accumulation in tumor and lymphoid compartments in all the tested mice (Fig. 7c–e). Conversely, the use of SAProsomes substantially increased the MSA-2 concentration in these tumors to 19.0–45.1-fold higher than that found after intravenous injection of free MSA-2, thus validating our design rationale of using drug chemical derivatization and liposome-based delivery. As an insight into the quantitative differences between SAProsomes and oral MSA-2 treatments, we observed a 22.0–449.1-fold higher drug accumulation in tumors following SAProsome-3 delivery, which contributed to the markedly improved antitumor efficacy of MSA-2. On the other hand, SAProsome-3 had low distribution across other normal organs, whereas intravenously administered MSA-2 accumulated at a higher concentration in the heart and kidneys (Supplementary Fig. S25).

## Nanoliposomal vesicles abrogate STING agonist-associated systemic toxicities

Finally, we investigated the potential toxicity induced by systemic administration of STING agonists. The same administration schedule with a 1.5-fold higher dose than therapeutically relevant doses used in the LLC model was tested. Serum biochemical analysis showed that following SAProsome-3 treatment, all parameters remained unchanged as compared to the healthy group (Fig. 7f, g). Conversely, the

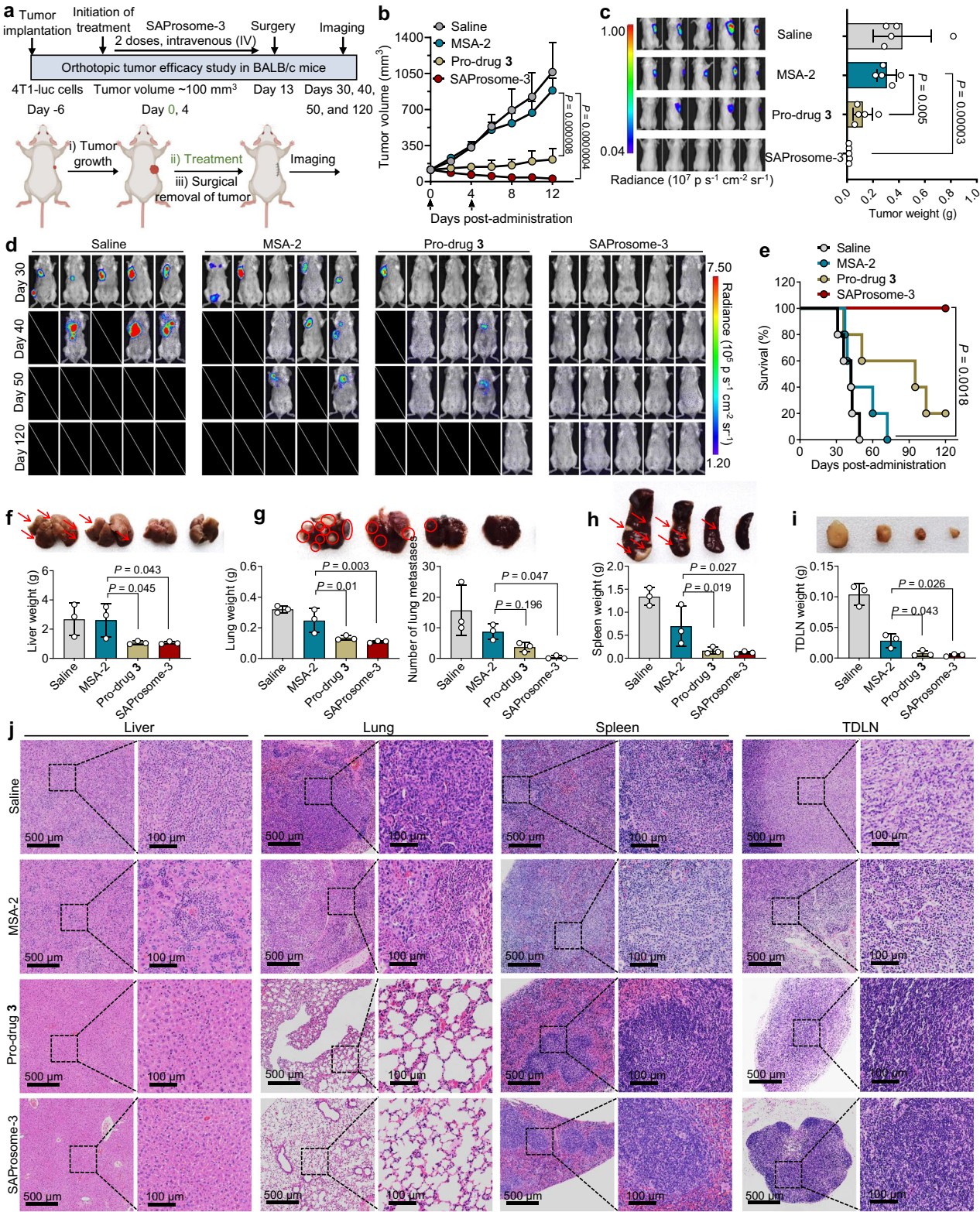

animals receiving intravenous MSA-2 experienced significant increases in aspartate aminotransferase (AST) and alanine aminotransferase (ALT), indicating obvious liver toxicities. Moreover, oral administration of MSA-2 also resulted in significantly elevated serum enzyme levels, including AST, ALT, urine acid (UA) and blood urea nitrogen (BUN). Histological examination of organ sections also suggested obvious toxicities to liver and kidney tissues after treatment with free MSA-2 via either intravenous or oral administration, while those

treated with SAProsome-3 were indistinguishable from untreated animals (Supplementary Fig. S26). We found acute inflammation in free drug-treated groups, characterized by elevated levels of inflammatory cytokines/chemokines (Fig. 7h and Supplementary Fig. S27). Of note, interleukin 6 (IL-6, a key driver of cytokine storm) maintained a 12.5- or 18.6-fold higher level in serum from mice treated with MSA-2 orally or intravenously compared to that of SAProsome-3 (Fig. 7h). In contrast, SAProsome-3 showed negligible fluctuation of these cytokines/

**Fig. 5 | SAProsome effectively inhibited tumor metastasis and recurrence after surgery. a** Schematic illustration of STING-activating immunotherapy for the treatment of spontaneous tumor metastases in the orthotopic 4T1 breast tumor model. 4T1-luc cells ($2.0 \times 10^5$) were inoculated into the mammary fat pads of syngeneic BALB/c mice. When 4T1 tumors reached ~100 mm³ in volume, mice were treated with saline (intravenous injection), free MSA-2 (oral administration), pro-drug **3** (oral administration), or SAProsome-3 (intravenous injection) on days 0 and 4. On day 12, the primary orthotopic tumors were surgically resected in each mouse for further tracking the metastases of 4T1-luc cancer cells. Created with BioRender.com. **b** Primary tumor-growth curves of 4T1 tumor-bearing mice in different treatment groups (n = 5 mice/group). Data are presented as the mean ± s.d. and statistical significance was determined using one-way analysis of variance (ANOVA). **c** In vivo bioluminescence images of tumor-bearing mice (left panel) before surgery and excised primary tumor weights (right panel). n = 5 biologically independent

animals. Data are presented as the mean ± s.d. and statistical significance was determined using one-way ANOVA. **d** Whole-body bioluminescence imaging to track the tumor burden in various drug-treated mice. **e** Mouse survival was monitored over the course of 120 days after surgery (n = 5 mice/group, log-rank Mantel–Cox test). **f–i** Weights of organs, including those of the liver (**f**), lung (**g**), spleen (**h**), and tumor-draining lymph nodes (TDLN) (**i**) in each treatment group. Mice were euthanized on day 30 for organ excision (n = 3 mice/group). Inset: representative photographs of harvested organs. Numbers of metastatic foci in lungs were counted (**g**). Red arrows or circles indicate tumor metastases. Data are presented as the mean ± s.d. and statistical significance was calculated by one-way ANOVA. **j** Histological analysis of organ sections using H&E staining to examine metastasis. The images on the right panel are the enlargement of the region in the black rectangle. Triplicates were performed independently with similar results. Source data are provided as a Source data file.

chemokines in serum and healthy organs, indicating minimized systemic inflammation and immunotoxicities.

## Discussion

Immunotherapy has tremendously impacted the treatment of a broad range of cancers, including metastatic cancer. However, clinically approved ICB monotherapy in unselected populations of patients with cancer demonstrated reduced response rates. This unfavorable response is due to several factors, such as a low number of immunogenic neoantigens, multiple immunosuppressive features in the TME, and paucity of tumor-infiltrating lymphocytes[44]. This provoked efforts consequently to develop novel treatment options to combat ICB-nonresponsive cancer. Priming innate immunity potentially generates greater antitumor adaptive immunity, which induces increased tumor-infiltrating lymphocytes and stronger treatment responses[45]. As an innate immune regulator, the STING protein is a critical sensor of self- and pathogen-derived DNA molecules, triggering numerous host defense pathways against viruses or bacteria. Furthermore, STING-mediated immunity can be exploited for the development of anti-tumor immunotherapies[46]. STING activation via agonists triggers downstream signaling events, engendering increased secretion of cytokines, including of type-I IFNs, which are essential for immunotherapeutic effects. Recently, substantial effort has been devoted to the discovery of STING agonists and various sophisticated delivery vehicles as antitumor therapeutics[47–50]. CDN-based agonists are hydrophilic and negatively charged, nanoparticle formulation and systemic delivery of these polar agents for therapeutic purposes remain challenging. In addition, the scalable synthesis of cGAMP analogs has posed substantial obstacles, requiring a tedious and lengthy synthetic protocol due to their functionally dense, polar structures[51]. In contrast, the synthetic strategy for these non-nucleotide MSA-2 pro-drugs described herein is straightforward and can be readily produced on a gram scale from commercially available reagents. Hence, we can expect these compounds could be of significant interest for further clinical translation.

The oral administration of MSA-2 only achieved low therapeutic responses in animal models, necessitating a higher dose for treatment[24]. Our initial attempts at encapsulating MSA-2 into state-of-the-art nanocarriers, such as liposomes or polymeric micelles, failed because of the low miscibility of MSA-2 with these matrices. We hypothesized that the carboxyl moiety (10-OH) on the MSA-2 molecule may account for this incompatibility. This motivated us to rationally re-engineer this agonist for adaptive nanoparticle formulation. We initially exploited a hydrophobic tocopherol motif to construct an MSA-2 derivative (i.e., pro-drug **5**) (Supplementary Fig. S28). Unfortunately, the resulting pro-drug **5** failed to promote DC maturation or trigger type-I IFN secretion. The hydrolysis rate of the ester bond in this pro-drug scaffold was slow, which inhibited the spontaneous release of active MSA-2 after uptake by cells. Furthermore, compared with orally administered free MSA-2, the tocopherol-derived STING pro-drug

delivered via liposomes failed to yield additional therapeutic advantages following intravenous injection (Supplementary Fig. S29). These results suggest a high intracellular concentration threshold for STING activation and sustained and slow activation of MSA-2 attenuates its efficiency as a STING activator. These preliminary findings prompted us to hypothesize that robust immune responses may be elicited by improving drug activation efficiencies via the optimization of linker chemistry and overall pro-drug entities.

Inspired by this hypothesis, we exploited MP-typed alkanols with varying hydrocarbon chain lengths as promoieties to chemically re-engineer the MSA-2 compound. The resultant pro-drugs were expected to exhibit lower hydrophobicity, less steric hindrance, and more accessibility for esterase hydrolysis than found with the tocopherol-derived pro-drug. Remarkably, we found that these pro-drugs were conducive to esterase-catalyzed bond cleavage, spontaneously regenerating biologically active drugs. Interestingly, we also found that drug activation kinetics was dependent on carbon numbers in the pro-drug linker. Compared with compound **5**, accelerated release profiles were observed for these pro-drugs (Supplementary Fig. S30); for example, ~90% of pro-drugs **3** or **4** were converted into free MSA-2 within 30 min in the presence of esterase. The rapid esterase-catalyzed conversion rate benefited their ability to activate STING and thereby initiate the downstream production of type-I IFNs, as evidenced by various in vitro cell-based assays. Notably, compared with the substantial challenges posed by the scalable synthesis of cGAMP analogs, the synthesis protocol for the agonist pro-drugs presented here is straightforward. These derivatives can be readily produced on a gram scale from commercially available reagents via a two-step reaction protocol, and hence, are beneficial for facilitating further clinical translation.

Liposomes represent the most promising drug delivery carriers for clinical use[52]. Currently, numerous liposome formulations are available on the market for disease treatment, with more formulations entering preclinical and clinical development. Contrary to the immiscibility of free MSA-2 with lipid constituents, the STING pro-drug can be stably constrained into liposomal vesicles to enable intravenous injection for in vivo investigation. The intravenous administration of STING-activating liposomal vesicles substantially improved pharmacokinetic properties and favorable delivery efficiencies of MSA-2 in the tumors or lymph nodes of multiple tumor models compared with that found with free MSA-2 administration. Notably, SAProsome-3 outperformed the other SAProsomes. We disclosed that the overall chemical structures of pro-drugs affected the hydrolysis kinetics and activation of MSA-2 in response to esterase. In fact, a positive correlation between hydrolysis rate and STING-stimulating activity was observed in this study. Moreover, SAProsome-3 had the highest $AUC_{0-t}$, which could, in part, explain its superior antitumor efficacy. Despite these findings, the structure-activity relationships remain to be explored in depth. Furthermore, consistent with these results, intravenously

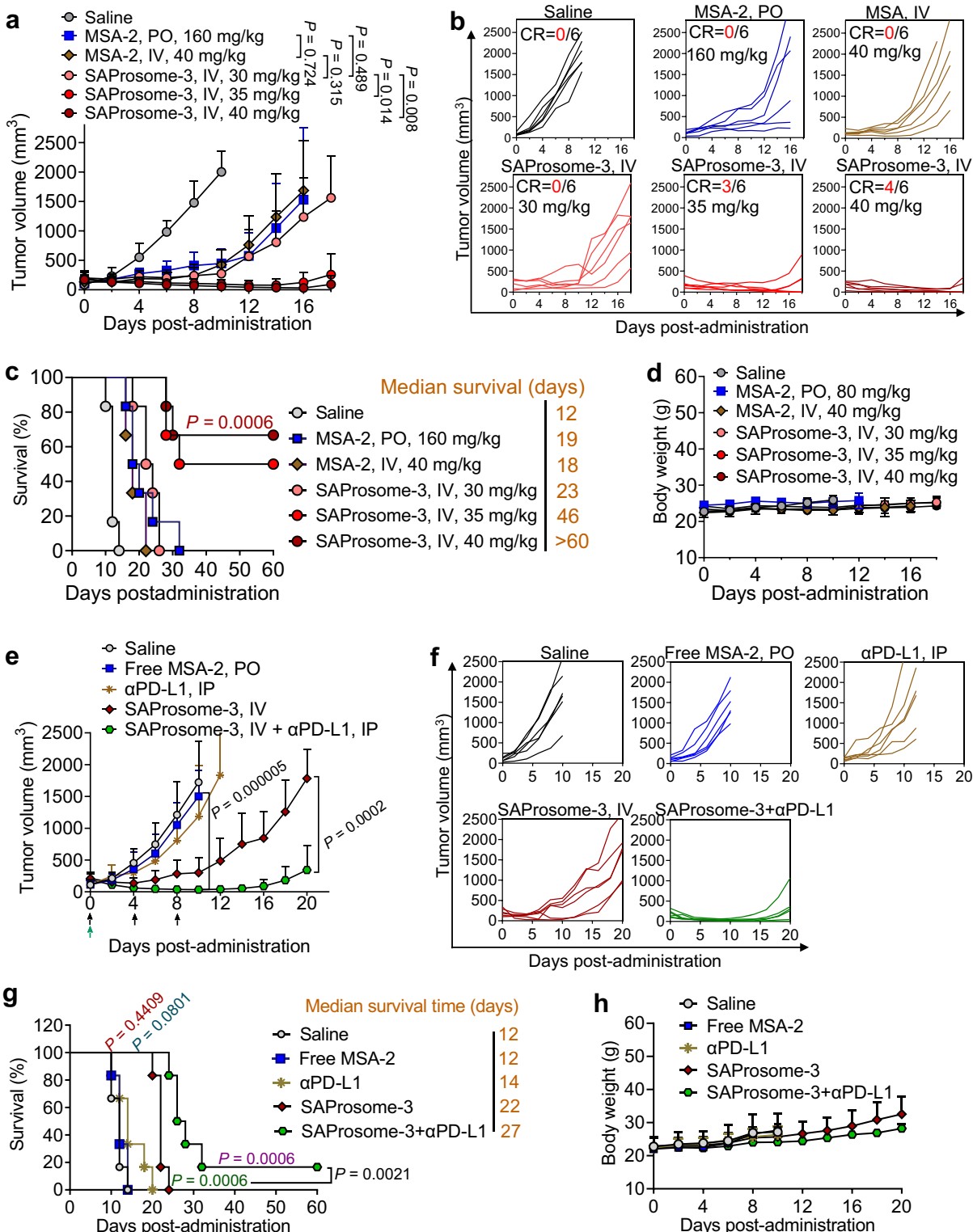

**Fig. 6 | SAProsome-3 potentiates therapeutic efficacy and synergizes with checkpoint blockade inhibition. a**–**d** LLC tumor-bearing mice were treated with free MSA-2 via intravenous (IV) administration or oral gavage, and SAProsome-3 following intravenous injection at indicated MSA-2–equivalent doses. Tumor growth curves (**a**), spider plots of individual tumor growth curves (**b**), Kaplan–Meier survival curves (two-sided log-rank test) (**c**) and body weight changes (**d**) of tumor-bearing mice are shown (*n* = 6 mice/group). **e**–**h** Mice with ~150 mm³ subcutaneous B16F10 tumors were administered with a single dose of saline (IV), free MSA-2 (PO) or SAProsome3 (IV) equivalent to 35 mg/kg MSA-2. For combined treatment with the checkpoint blockade antibody, mice were additionally administered with anti-PD-L1 antibody intraperitoneally three times at a dose of 5 mg/kg (*n* = 6 mice/group). **f** Tumor-growth curves for individual mice. **g** Kaplan–Meier survival analysis (two-sided log-rank test). The mice were treated with the indicated formulations, and a tumor volume of 2000 mm³ was set as the endpoint criteria. **h** Body weight change of B16F10 tumor-bearing mice during treatment (*n* = 6 mice/group). In (**a**, **d**, **e**, **h**), data are presented as the mean ± s.d. and statistically analyzed using one-way analysis of variance. Source data are provided as a Source data file.

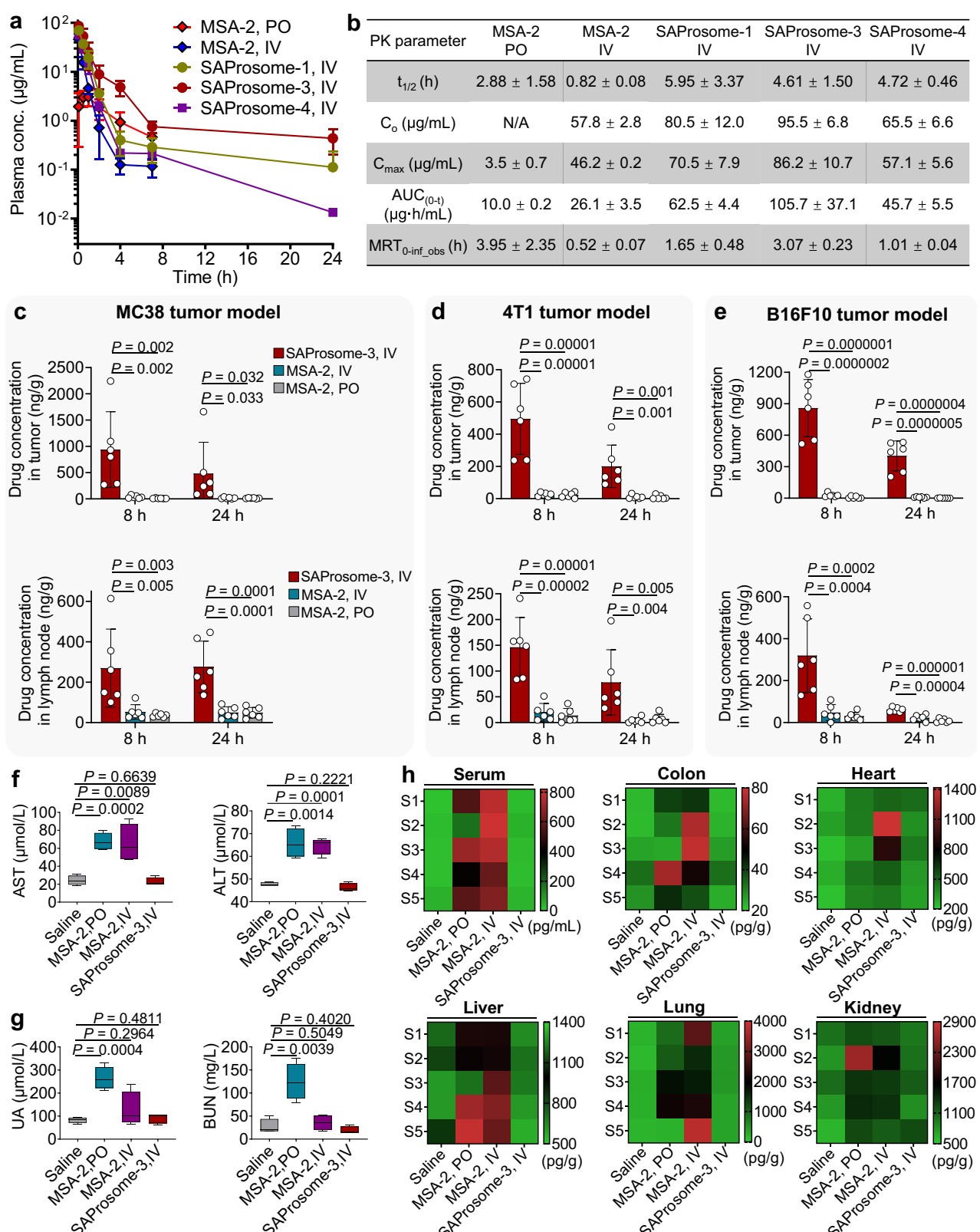

administered SAProsome-3 provoked vigorous tumor remission in MC38 and LLC murine models and enabled the complete eradication of established tumors. Mechanistically, SAProsome-3 treatment promoted a type-I IFN-driven innate immune response, thereby transforming immunologically cold tumors into hot ones. Following SAProsome-3 treatment, we indeed observed increased tumor-infiltration by functional T cells, NK cells, and activated DCs

as well as a change in the polarization of macrophages from pro-tumorigenic M2 to antitumorigenic M1 phenotypes, all of which contributed to a robust antitumor immune response in this MC38 model. In particular, in 4T1 and B16F10 tumors, which are regarded as immunologically silent and not amenable to ICB monotherapy, the intravenous administration of the optimized SAProsome-3 effectively suppressed tumor growth and metastatic burden,

**Fig. 7 | Pharmacokinetic properties, biodistribution, and safety profiles of SAProsomes. a** Drug concentration in the plasma of Sprague-Dawley rats ($n = 3$ rats/group) following a single administration of different drugs at a 17.5 mg/kg of MSA-2 equivalent dose. Data are presented as ± s.d. of the mean. **b** Pharmacokinetic parameters reflecting half-life ($t_{1/2}$), $C_0$, $C_{max}$, area under the curve ($AUC_{0-t}$) and mean residence time (MRT). Mice bearing MC38 (**c**), 4T1 (**d**), or B16F10 (**e**) xenograft tumors were given a single dose of MSA-2 and SAProsome-3 ($n = 6$ mice/group). Drug concentration in tumors and lymph nodes was determined using HPLC analysis at 8 and 24 h after administration. Data are presented as the mean ± s.d. and statistically analyzed using one-way analysis of variance.

**f, g** C57BL/6 mice were treated with free MSA-2 orally at a dose of 240 mg/kg or intravenously at a dose of 60 mg/kg, and SAProsome-3 intravenously at a dose of 45 mg/kg per mouse, every three days for a total of three injections ($n = 4$ mice/group). Blood chemistry of mice were analyzed on day 8 postadministration. Box plot conveys median (middle line), 25th and 75th percentiles (box), and the minima to maxima range (whiskers). Data are statistically analyzed using one-way analysis of variance. **h** The levels of interleukin 6 (IL-6) in the serum and other tissues were determined using ELISA at 6 h postadministration of free MSA-2 (PO, 240 mg/kg), free MSA-2 (IV, 60 mg/kg) or SAProsome-3 (IV, 45 mg/kg) ($n = 5$). Source data are provided as a Source data file.

further synergizing with αPD-L1 immune checkpoint inhibition to enhance therapeutic benefit.

Although the intratumoral delivery of STING agonists may mitigate systemic toxicities, this administration route is restricted to patients with solid accessible tumors. Our experimental results provide a proof-of-concept for low-efficiency STING agonists being repurposed and structurally optimized for generating systemically injectable, high-efficacy nanotherapeutics to address their immune escape mechanisms. Furthermore, given the significantly improved oral activity of pro-drug **3** observed in the experiments, a prompt efficacy evaluation of using this compound via oral administration can be envisioned. Following safety assessment, prospective clinical trials for investigating the optimized STING-activating liposomes alone or in combination with other immunotherapies are warranted.

## Methods

### Ethical statement
This research complies with all relevant ethical regulations. All animal studies were conducted in accordance with the National Institute Guide for the Care and Use of Laboratory Animals. The experimental protocols were approved by the Ethics Committee of the First Affiliated Hospital, Zhejiang University School of Medicine.

### Animal study
C57BL/6 (male, 4–6 weeks, 18–20 g, #ZJCLA-202004), BALB/c mice (female, 4–6 weeks, 15–18 g, #ZJCLA-202101) and Sprague-Dawley rats (male, 6–8 weeks, 200–220 g, #ZJCLA-202003) were purchased from the Laboratory Animal Center of Hangzhou Medical College (Hangzhou, China). Animals were raised in specific pathogen-free animal experimental center and allowed free access to food and water. All experimental/control animals were co-housed in a habitant under standard conditions (23–26 °C, 40–60% humidity, 12 h light–dark cycle, and 3–4 mice or rats/cage). At the end point of the study, animal euthanasia was performed by $CO_2$ inhalation followed by cervical dislocation.

### Synthesis of MSA-2–based pro-drugs
The chemical structures of the synthesized MSA-2 pro-drugs were characterized using proton nuclear magnetic resonance and high-resolution mass spectrometry. The synthetic procedures are described in detail in the Supplementary Materials and Methods.

### Preparation and characterization of liposomal STING agonists
SAProsomes were prepared according to an ethanol dilution method, as described previously. Briefly, first, a lipid mixture of egg-phosphatidylcholine, cholesterol, and 1,2-distearoyl-sn-glycero-3-phosphoethanolamine-N-poly(ethylene glycol)$_{2000}$ was prepared in ethanol (0.9 mL) at a mass ratio of 7:2:1. Next, 0.1 mL of dimethylsulfoxide containing MSA-2 pro-drug (20 mg/mL, equivalent to MSA-2) was added to the above ethanolic solution in the ratio of lipids to MSA-2 fixed at 14:1 (w/w). The mixture (1 mL) was then rapidly injected into PBS (9 mL, pH = 7.4), yielding stable liposomes in PBS with an MSA-2–equivalent concentration of 0.2 mg/mL. Finally, the resulting SAProsomes were reprecipitated at $100,000 \times g$ for 30 min via ultracentrifugation (Berkman, Optima™ L-100 XP Ultracentrifuge) and washed with PBS thrice to remove organic solvents. The drug concentration was determined using analytical high-performance liquid chromatography (HPLC).

### Cell lines and cell culture
The human monocyte cell line THP1 cells and LLC cells were purchased from the Cell Bank of the Chinese Academy of Sciences (Shanghai, China) and were cultured in Roswell Park Memorial Institute 1640 medium supplemented with 10% fetal bovine serum, 100 units/mL of penicillin, and 100 μg/mL of streptomycin. B16F10-OVA cell line was purchased from Crisprbio (Beijing, China). Primary BMDC cultures were established from female C57BL/6 mice. Briefly, bone marrow was flushed from isolated femurs using PBS. The marrow was seeded in petri dishes containing Dulbecco's Modified Eagle Medium supplemented with 10% heat-inactivated fetal bovine serum, 1% penicillin–streptomycin, 5 ng/mL interleukin-4, and 20 ng/mL murine granulocyte macrophage colony-stimulating factor. Half the culture medium volume was refreshed on day 3. The nonadherent cell fractions of these cultures (typically >75% CD11c$^+$ using flow cytometry) were used for experiments between days 6 and 8 of culture. All cells were cultured in a humid atmosphere at 37 °C and 5% $CO_2$.

### In vitro evaluation of cellular uptake
To quantify the relative cellular MSA-2 uptake, BMDCs and THP1 cells were plated ($4 \times 10^5$/per well) in a 24-well plate, treated with the indicated formulations at 40 μM of MSA-2, and then suspended in PBS and analyzed via flow cytometry (DxFLEX, Beckman Coulter) using a 405-nm excitation laser and 450/45-nm filter configuration.

### Tumor therapy experiments
Mice were subcutaneously inoculated with $5 \times 10^6$ MC38 cells into the right flank. Tumor size was measured every 2 days using a digital caliper, and tumor volume was calculated as $0.5 \times \text{length} \times \text{width}^2$. On reaching sizes of ~100 mm³, the animals were randomized into six groups ($n = 10$ in each group). Three injections of different SAProsomes were intravenously administered via the tail vein at an MSA-2 dose of 35 mg/kg on days 0, 4, and 8, while free MSA-2 was orally administrated and saline was intravenously injected as controls. Tumor growth and body weight were monitored and recorded every 2 days. The mice were euthanized in accordance with the animal ethics guidelines of our institute and animal welfare regulations when the tumor volume reached 2000 mm³. Mice were euthanized via $CO_2$ inhalation at the end point of the study.

### Histological analysis and immunostaining
For histological analysis, the excised tumors and organs were fixed in 4% paraformaldehyde in phosphate buffer, embedded in paraffin, and sectioned into 5-μm–thick slices. Tumor and organ sections were stained with hematoxylin and eosin (H&E). For the terminal deoxynucleotidyl transferase-mediated dUTP nick end labeling (TUNEL) assay, the dewaxed and rehydrated tumor sections were incubated with proteinase K for 15 min at 37 °C, rinsed with PBS twice, and rinsed

with the TUNEL in situ cell death detection kit according to the manufacturer's protocol (Sigma-Aldrich). The TUNEL-stained cells were counter-stained with diaminobenzidine (DAKO). To assess immune cell infiltration, samples were fixed and stained with antibodies (CD8a and Granzyme B, Biolegend) and DAPI. Slides were imaged using a fluorescence microscope (Olympus, IX71).

## In vivo pharmacokinetic properties of SAProsomes
The plasma concentrations of SAProsomes (at an MSA-2 equivalent dose of 17.5 mg/kg) administered intravenously were compared with those of free MSA-2 (17.5 mg/kg) administered orally or intravenously in Sprague-Dawley rats (~200 g, $n = 3$ in each group). Following drug treatment, blood samples were collected at predetermined time points and immediately centrifuged for 10 min at $3000 \times g$, and the resulting plasma samples were collected in microfuge tubes. MSA-2 and prodrug concentrations after complete hydrolysis were analyzed via reverse-phase HPLC.

## Statistical analysis
Prism software version 8.0 (GraphPad) was employed to perform all statistical analyses. Unless otherwise indicated, data are presented as ±standard error of the mean. Statistical significance between measurements was assessed using unpaired Student's $t$ test or one-way analysis of variance. Survival studies were analyzed using Kaplan–Meier plots.

## Reporting summary
Further information on research design is available in the Nature Portfolio Reporting Summary linked to this article.

## Data availability
All data needed to evaluate the conclusions in the paper are present in the paper and/or the Supplementary Materials. Source data are provided with this paper.

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

## Acknowledgements

This work was supported by the National Natural Science Foundation of China (Grants. 82273490, 82073296, and 81773193 to H.W.), Research Project of Jinan Microecological Biomedicine Shandong Laboratory (Grant JNL-2022010B to H.W.), and Zhejiang Provincial Natural Science Foundation of China (Grant LR19H160002 to H.W.). We thank the technical assistance from the Center of Cryo-Electron Microscopy (CCEM), Zhejiang University, for TEM analysis. We thank Zhaoxiaonan Lin from the Core Facilities, Zhejiang University School of Medicine for her technical support.

## Author contributions

H.W. conceived and designed research; Xiaona Chen performed most experiments; F.M., Y.X., and Xiaolong Chen helped to perform research; Xiaona Chen, F.M., Y.X., Xiaolong Chen, T.L., and H.W. analyzed data; Xiaona Chen and H.W. wrote and revised the manuscript.

## Competing interests

The authors declare no competing interests.
