## [Peer Review File · Nature Communications]

Chemically programmed STING-activating nano-liposomal vesicles improve anticancer immunityREVIEWER COMMENTS

Reviewer #1 (Remarks to the Author):

This manuscript by Chen et al. reports on the development of a strategy for systemic delivery of a non-nucleotide small molecule STING agonist, MSA-2, via encapsulation of an MSA-2-prodrug in liposomal vehicles. The authors test different liposomal compositions and identify one, Saprosome-3, which is shown to be safe and effective in concentrating MSA-2 in tumors, leading to impressive anti-tumor activity in models of primary tumor treatment and prevention of metastatic recurrence, accompanied by induction of protective memory. Overall this strategy appears promising and the prodrug+liposome formulation is elegant.

I do have one major issue with the data as presented– the comparison to per-oral dosing. The original publication describing MSA-2 reported that it was safe when dosed to high levels orally or subcutaneously, but oral dosing at 20 mg/kg 3X was completely ineffective in tumor control while oral dosing at 80 mg/kg 3X elicited strong anti-tumor activity. The oral dosing used here is not completely clear– the liposomes are dosed at 35 mg/kg (MSA-2 equivalent dose), and it seems the authors may be using the same dose for oral free MSA-2. But this is not the comparison that should be made, as 35 mg/kg may be too low for efficacy when given orally—the liposomes at their MTD should be compared to free oral MSA-2 at its previously reported effective dose (i.e., order of 80 mg/kg). For clinical translation, it would be more attractive to give free MSA-2 at a higher dose orally than having to resort to i.v. administration, if the oral route gives the same tumor response without toxicity. There is no obvious benefit to the liposomes if their main impact is to allow a lower dose to be effective, if there aren't dose-limiting side effects of oral dosing at a higher dose.

Other main issues:

1. Line 157, the authors write: "SAProsome treatment engendered complete tumor regression (CR), and the tumor inhibition rates were highly dependent on the encapsulated MSA-2 prodrugs." Control treatment with empty liposomes is needed to confirm that the liposome doses used to achieve these high drug doses (35 mg/kg) are not directly having an anti-tumor effect.
2. There is substantial recent prior work on formulating STING agonists of different types for enhanced delivery to tumors, and the authors relegate this work to a single sentence citing one review on the field (ref 47). The authors largely dismiss this other work, indicating that "...as CDN-based small-molecule agonists are hydrophilic and negatively charged, the therapeutic delivery of these polar agents remains technically challenging." Yet here, MSA-2 also required synthesis of a prodrug and formulation in liposomes for effective delivery, which suggests this compound's delivery is equally technically challenging. A deeper discussion of the pros and cons of the current approach vs. other publications is needed.

Minor comments:

1. Line 134: The authors present data on cytokine and chemokine expression from DCs in vitro, and then state "These data indicate that the SAProsome platform can facilitate tumor-infiltration and activation of CD8+ T-effector cells." This is an overstatement, one can't claim this from these in vitro cytokine/chemokine secretion data.
2. Line 201: Text calls out Fig. 5I and J, this should be Fig. 4I and J.

Reviewer #2 (Remarks to the Author):

Summary: The manuscript by Chen et al. describes a nanoparticle-based strategy for systemic (i.v.) delivery of STING agonist prodrugs based on the MSA-2 small molecule STING agonist. The authors use the carboxylic acid of MSA-2, which is critical for STING binding, to generate a small library of lipophilic ester-based prodrugs that are then encapsulated into a liposomal formulation for systemic delivery (referred to as "prosomes"). Consistent with what is known about other

prodrugs that must be cleaved by esterases to generate the active agent, they find that the rate of conversion from prodrug to MSA-2 is dependent on carbon spacer length. They next demonstrate that the prodrugs, which become highly lipophilic, can be loaded into liposomal nanoparticles and that enhance STING activation in cell culture models compared to the MSA-2 parent compound. The authors then test the efficacy of 4 prosome formulations injected intravenously in an MC38 tumor model, demonstrating enhanced activity relative to MSA-2 that is administered orally and also identify a lead formulation (SAProsome-3) based on its potent antitumor activity. The authors then progress to evaluating how the lead formulation impacts the tumor microenvironment, demonstrating enhanced infiltration of CD8+ T cells and NK cells as well as a shift towards an increased number of activated DCs and M1-like microphages. The authors also test the efficacy of the lead formulation in a more challenging tumor models of 4T1 breast cancer metastasis and B16F10 melanoma. Finally, the authors evaluate the PK and biodistribution, finding that the liposomal formulation extends half-life over the free MSA-2 and increases tumor and lymph node accumulation of the STING agonist.

Overall evaluation: Overall, this is a solid manuscript that follows a logical workflow for development and testing of a new immunotherapy formulation and is also supported by strong data in multiple mouse tumor models that demonstrates potent efficacy of the platform. Unfortunately, the manuscript suffers from a major flaw in that it compares the i.v. administered prodrug formulations only to free MSA-2 that is administered orally. This is not a fair comparison as it well-established that the bioavailability of most drugs delivered orally is lower than that delivered i.v. There should not be a technical reason for not comparing i.v. MSA-2 to the prodrug/prosome formulations since i.v. delivery was used in their PK studies and there are a number of ways in which hydrophobic agents can be solubilized for i.v. delivery (for example, use of PEG or other excipients). At minimum, a more appropriate comparison would have been to subQ administration of free MSA-2 at 50 mg/kg as was performed in the original Pan et al. paper where they observed robust responses using the subQ route. While there are several other major questions that need to be addressed that are outlined below, a significant flaw of the current work is the lack of a comparison to free MSA-2 that is also delivered i.v., ideally at the MTD of the free agent but at minimum at the same dose. If such a comparison can be made and the data clearly show that the prodrug/liposomal formulations increases the efficacy or potency of MSA-2 then the manuscript may be suitable for publication provided other points are also addressed.

Major Points:

1. Pertaining to above, the authors need to perform extensive additional experiments to compare the prosome formulations to free MSA-2 administered via i.v. route, or at minimum, via the subQ route used by Pan et al, which also demonstrated enhanced efficacy compared to oral administration. The authors perform i.v. administration in their PK/biodistribution study and so this delivery route is feasible. Without such direct comparison and a critical control, conclusions regarding the relative efficacy of prosome compared to the free active agent cannot be made and the impact of the technology cannot be assessed.
2. Another important control is MSA-2 loaded into liposomes; i.e., not a prodrug form. The authors claim that they were not able to load MSA-2 into liposomes and it is logical that converting into the prodrug format would increase lipophilicity and loading into liposomes. However, the pKa of the carboxylic acid of MSA-2 is 4.8 and therefore it will also be water insoluble at lower pH values and so presumably would load well into liposomes under these conditions. Have the authors attempted such alternatives for loading of MSA-2 into liposomes so that a comparison between a liposomal formulation and a prosome can be made? Such a study is important for determining the relative importance of the prodrug feature relative to the liposomal delivery system.
3. The authors claim that uncontrolled systemic inflammation is an issue for systemically administered STING agonists and that their design might mitigate this problem. However, the authors do not quantify levels of serum cytokines or the levels of STING-driven inflammation in major distribution organs and so the relative impact of the design on systemic inflammation compared to delivery of a free MSA-2 agonist is not known. These experiments need to be performed to help determine if the design can minimize systemic inflammatory effects.
4. Related to above, while the weight loss data provided suggests that the system is well-tolerated, additional analysis of toxicity, including serum markers of liver/kidney damage as well as organ pathology should be performed to clearly demonstrate relative toxicity of the system. Also, the body weight data (Fig 3C) should instead be plotted as % loss or gain from original mass.

5. Considering that the rest of the studies, including biodistribution studies, were performed in mice, it was not clear why the pharmacokinetic analysis was done in rats. It would have been much more instructive to do this in mice so that relationships between pharmacological properties, efficacy, and toxicity can be drawn.
6. Related to above, the authors also need to evaluate the biodistribution of drug in other distribution organs such as liver, spleen, and potentially lung; only tumor and LNs are shown and, while these are important, the ratio of drug in these tissues relative to that in other organs is more important. Further, in the original Pan et al. paper it is shown that free MSA-2 preferentially accumulates in tumor tissue due to change in solubility due to protonation of the carboxy group within the tumor microenvironment. So, an evaluation of drug distribution to other organs and the ratio of drug in tumor/other tissues needs to be performed.
7. In Figure 2C, the activity of prosomes vs. free MSA2 are shown at 2 different concentrations. First, this (or similar) should be done over a wider range of concentrations and an EC50 estimated for the different groups; this will allow for the relative potencies of each to be determined, which is an important metric when comparing new formulations. Second, statistical analysis of this data is missing.
8. In Figure 2I, a control should be run using either non-Ova-pulsed DCs and/or B16 cells that do not express Ova in order to demonstrate that the killing response is antigen specific. It is somewhat surprising that there would be a sufficient number of Ova-specific T cells in a wild-type mouse that can be expanded in vitro to kill target cells. Therefore, it is possible that what is being observed is non-specific target cell killing; the controls mentioned above are necessary to demonstrate this.
9. A technical consideration is why the percentage of CD86+CD80+ DCs is so high (~25%) in the untreated group (Figure 2G); this may suggest a need to better titrate antibodies or perhaps another source of innate immune activation (e.g., cell stress, contaminants in cell culture).
10. A challenge with ester-based prodrugs is potential to be cleaved by murine esterases but not human esterases. The authors should confirm prodrug activity in human cells as well.
11. A more minor point is that the authors are encouraged to expand upon their discussion as to why prosome3 seems to have the best therapeutic efficacy of the 4 tested. This is an interesting finding and so the authors should offer some reasons as to why this might be the case.

Reply to comments of Reviewer #1 (Remarks to the Author):

This manuscript by Chen et al. reports on the development of a strategy for systemic delivery of a non-nucleotide small molecule STING agonist, MSA-2, via encapsulation of an MSA-2-prodrug in liposomal vehicles. The authors test different liposomal compositions and identify one, Saprosume-3, which is shown to be safe and effective in concentrating MSA-2 in tumors, leading to impressive anti-tumor activity in models of primary tumor treatment and prevention of metastatic recurrence, accompanied by induction of protective memory. Overall this strategy appears promising and the prodrug+liposome formulation is elegant.

I do have one major issue with the data as presented—the comparison to per-oral dosing. The original publication describing MSA-2 reported that it was safe when dosed to high levels orally or subcutaneously, but oral dosing at 20 mg/kg 3X was completely ineffective in tumor control while oral dosing at 80 mg/kg 3X elicited strong anti-tumor activity. The oral dosing used here is not completely clear- the liposomes are dosed at 35 mg/kg (MSA-2 equivalent dose), and it seems the authors may be using the same dose for oral free MSA-2. But this is not the comparison that should be made, as 35 mg/kg may be too low for efficacy when given orally—the liposomes at their MTD should be compared to free oral MSA-2 at its previously reported effective dose (i.e., order of 80 mg/kg). For clinical translation, it would be more attractive to give free MSA-2 at a higher dose orally than having to resort to i.v. administration, if the oral route gives the same tumor response without toxicity. There is no obvious benefit to the liposomes if their main impact is to allow a lower dose to be effective, if there aren't dose-limiting side effects of oral dosing at a higher dose.

Response:

We would like to express our sincere appreciation to the reviewer for taking the time to review our manuscript and for providing positive feedback on our work. We also appreciate the reviewer for these useful comments made on our manuscript. In the original submission, we focused the studies on the efficacy testing at the same doses of MSA-2 in different tumor models, while having ignored the toxicity evaluation. We agree with the reviewer the opinion that it would be more attractive to give free MSA-2 at a higher dose orally. We therefore compared the antitumor activity of different administration routes to at different doses and sought to determine the doses of each formulation that have equal antitumor efficacy.

For this purpose, we conducted the efficacy studies against a Lewis lung carcinoma (LLC) model. We observed the dose-dependent antitumor activity for each MSA-2 formulation and

administration route. Interestingly, low doses of SAProsome-3 such as 35 or 40 mg/kg potently suppressed the tumor growth in this LLC model. These results and relevant description are supplemented in the revised manuscript (Page 11-12) and **supplementary Figures S23 and S24** as follows.

Efficacy comparison of SAProsome-3 versus free MSA-2 via different routes

We next sought to compare the therapeutic efficacy of SAProsome-3 *versus* free MSA-2. The mouse tumor model bearing Lewis lung carcinoma (LLC) was established and the animals were treated with different doses and administration routes of therapeutics. Dose-dependent antitumor activity was observed in each treatment group (**Fig. 6a** and **Supplementary Fig. S23**). Interestingly, in this model, we still observed the striking potency of intravenous SAProsome-3, yielding durable tumor regression and complete responses in 50% and 66.7% of mice at 35 and 40 mg/kg of MSA-2–equivalent doses, respectively (**Fig. 6a** and **b**). Notably, dosing of SAProsome-3 at 40 mg/kg led to significant tumor volume shrinkage to ~30 mm³. In sharp contrast, free MSA-2 administered either by oral or intravenous route only showed limited efficacy against LLC tumors (**Fig. 6a-c** and **Supplementary Fig. S23**). We also confirmed the doses of oral MSA-2 at 160 mg/kg, intravenous MSA-2 at 40 mg/kg, and intravenous SAProsome-3 at 30 mg/kg (MSA-2 equivalence), in which they had the comparable antitumor efficacy in this animal model (**Fig. 6a-c**). Histological analysis using H&E staining and TUNEL-positive cells labeling revealed pronounced intratumoral apoptosis after SAProsome-3 treatment (**Supplementary Fig. S24**). Moreover, administration of SAProsome-3 enhanced infiltration of CD8⁺ T cells in the tumors, which correlated positively with elevated production of granzyme B (**Supplementary Fig. S24**). Again, SAProsome-3 was proven to be safe for intravenous administration as supported by stable body weights in animals (**Fig. 6d**).

Fig. 6. SAProsome-3 potentiates therapeutic efficacy and synergizes with checkpoint blockade inhibition. a–d LLC tumor-bearing mice were treated with free MSA-2 via intravenous (IV) administration or oral gavage, and SAProsome-3 following intravenous injection at indicated MSA-2–equivalent doses. Tumor growth curves **a**, spider plots of individual tumor growth curves **b**, Kaplan–Meier survival curves **c** and body weight changes

d of tumor-bearing mice are shown (n = 6 mice/group). **e–h** Mice with ~150 mm³ subcutaneous B16F10 tumors were administered with a single dose of saline (IV), free MSA-2 (PO) or SAProsome3 (IV) equivalent to 35 mg/kg MSA-2. For combined treatment with the checkpoint blockade antibody, mice were additionally administered with anti-PD-L1 antibody intraperitoneally three times at a dose of 5 mg/kg (n = 6 mice/group). **f** Tumor-growth curves for individual mice. **g** Kaplan–Meier survival analysis (log-rank test). The mice were treated with the indicated formulations, and a tumor volume of 2000 mm³ was set as the endpoint criteria. **h** Body weight change of B16F10 tumor-bearing mice during treatment (n = 6 mice/group).

We also agree with the reviewer that it would be more appealing to administer free MSA-2 at a higher dose orally rather than having to resort to intravenous administration, if the oral route gives the same tumor response without toxicity. According to this comment, we investigated the potential toxicity induced by oral or intravenous administration of free MSA-2, which was compared with intravenously administered SAProsome-3. In the LLC model, we have confirmed that oral MSA-2 at 160 mg/kg, intravenous MSA-2 at 40 mg/kg, and intravenous SAProsome-3 at 30 mg/kg (MSA-2 equivalence) showed comparable antitumor activity (**Fig. 6a-c**). Hence, we assessed the toxicity of drugs at 1.5-fold higher doses than the abovementioned therapeutically relevant doses.

As the reviewer notes, although free MSA-2 taken orally was able to induce antitumor activity, a higher dose was required, which may cause dose-limiting side effects in animals. The results of *in vivo* toxicity studies showed that free drug treatment induced acute inflammation, characterized by elevated levels of inflammatory cytokines/chemokines in the blood and major organs. In addition, hepatic and renal toxicities were also observed in free MSA-2–treated mice. In sharp contrast, SAProsome-3 had negligible *in vivo* toxicities such as liver and renal damages and system inflammation when administered at 45 mg/kg (MSA-2 equivalence). Therefore, we expected that the SAProsome scaffold could optimize the *in vivo* delivery of MSA-2 agonist against cancer, triggering substantial antitumor immune responses within the safe dosing range. We have included these results in the revised manuscript (Page 14-15) shown as below.

Nanoliposomal vesicles abrogate STING agonist-associated systemic toxicities

Finally, we investigated the potential toxicity induced by systemic administration of STING agonists. The same administration schedule with a 1.5-fold higher dose than therapeutically

relevant doses used in the LLC model was tested. Serum biochemical analysis showed that following SAProsome-3 treatment, all parameters remained unchanged as compared to the healthy group (**Fig. 7f-g**). Conversely, the animals receiving intravenous MSA-2 experienced significant increases in aspartate aminotransferase (AST) and alanine aminotransferase (ALT), indicating obvious liver toxicities. Moreover, oral administration of MSA-2 also resulted in significantly elevated serum enzyme levels, including AST, ALT, urine acid (UA) and blood urea nitrogen (BUN). Histological examination of organ sections also suggested obvious toxicities to liver and kidney tissues after treatment with free MSA-2 via either intravenous or oral administration, while those treated with SAProsome-3 were indistinguishable from untreated animals (**Supplementary Fig. S26**). We found acute inflammation in free drug-treated groups, characterized by elevated levels of inflammatory cytokines/chemokines (**Fig. 7h** and **Supplementary Fig. S27**). Of note, IL-6 (a key driver of cytokine storm) maintained a 12.5 or 18.6-fold higher level in serum from mice treated with MSA-2 orally or intravenously compared to that of SAProsome-3 (**Fig. 7h**). In contrast, SAProsome-3 showed negligible fluctuation of these cytokines/chemokines in serum and healthy organs, indicating minimized systemic inflammation and immunotoxicities.

Fig. 7. Pharmacokinetic properties, biodistribution and safety profiles of SAProsomes.
a Drug concentration in the plasma of Sprague-Dawley rats ($n = 3$ rats/group) following a single administration of different drugs at a 17.5 mg/kg of MSA-2 equivalent dose. **b** Pharmacokinetic parameters reflecting half-life ($t_{1/2}$), C_0 , C_{max} , area under the curve (AUC_{0-t}) and mean residence time (MRT). Mice bearing MC38 **c**, 4T1 **d**, or B16F10 **e** xenograft tumors

were given a single dose of MSA-2 and SAProsome-3. Drug concentration in tumors and lymph nodes was determined using HPLC analysis at 8 and 24 h after administration. **f-g** C57BL/6 mice were treated with free MSA-2 orally at a dose of 240 mg/kg or intravenously at a dose of 60 mg/kg, and SAProsome-3 intravenously at a dose of 45 mg/kg per mouse, every three days for a total of three injections. Blood chemistry of mice were analyzed on day 8 postadministration. **h** The levels of IL-6 in the serum and other tissues were determined using ELISA at 6 h postadministration of free MSA-2 (PO, 240 mg/kg), free MSA-2 (IV, 60 mg/kg) or SAProsome-3 (IV, 45 mg/kg) (n = 5).

Other main issues:

1. Line 157, the authors write: “SAProsome treatment engendered complete tumor regression (CR), and the tumor inhibition rates were highly dependent on the encapsulated MSA-2 prodrugs.” Control treatment with empty liposomes is needed to confirm that the liposome doses used to achieve these high drug doses (35 mg/kg) are not directly having an anti-tumor effect.

Response:

Thanks for this comment. We accordingly evaluated the antitumor effect of empty liposomes in the MC38 model. We have incorporated this result in the revised manuscript (Page 8) and included the data in **supplementary Figure S14**.

As a comparison, empty liposomes without prodrug payloads did not show any antitumor benefits in this model (**Supplementary Fig. S14**).

Fig. S14. Tumor growth curves of MC38 xenografts in mice. Mice were treated with either saline or empty liposomes without prodrug payloads.

2. There is substantial recent prior work on formulating STING agonists of different types for enhanced delivery to tumors, and the authors relegate this work to a single sentence citing one review on the field (ref 47). The authors largely dismiss this other work, indicating that "...as CDN-based small-molecule agonists are hydrophilic and negatively charged, the therapeutic delivery of these polar agents remains technically challenging." Yet here, MSA-2 also required synthesis of a prodrug and formulation in liposomes for effective delivery, which suggests this compound's delivery is equally technically challenging. A deeper discussion of the pros and cons of the current approach vs. other publications is needed.

Response:

We acknowledge that there is substantial prior work on formulation of STING agonists for better delivery to tumors. According to this comment, we have added more discussion on the advantages and limitations of our approach compared to other publications. We agree with the reviewer that our approach also requires the synthesis of a prodrug and its formulation in liposomes for effective delivery. However, compared with CDN-based STING agonists, the synthesis and preparation protocol for the MSA-2 prodrug and its liposomes is straightforward and simple. The MSA-2 prodrugs can be readily produced from commercially available reagents using a two-step reaction protocol, which is beneficial for future clinical translation. On the other hand, liposomes are the most promising drug delivery carriers for clinical use. We here found that the rationally engineered STING prodrugs can be stably incorporated into liposomal vesicles, enabling intravenous injection for *in vivo* studies. Compared to the immiscibility of free MSA-2 with lipid constituents, our approach enables better pharmacokinetic properties and efficient delivery of MSA-2 to tumors or lymph nodes in multiple tumor models. Hence, we can expect these compounds to exhibit a high potential for further clinical translation. We appreciate the reviewer's valuable feedback and have incorporated the relevant discussion into the revised manuscript (Page 15-16) as follows.

Recently, substantial effort has been devoted to the discovery of STING agonists and various sophisticated delivery vehicles as antitumor therapeutics⁴⁷⁻⁵⁰. CDN-based agonists are hydrophilic and negatively charged, nanoparticle formulation and systemic delivery of these polar agents for therapeutic purposes remain challenging. In addition, the scalable synthesis of cGAMP analogues has posed substantial obstacles, requiring a tedious and lengthy synthetic protocol due to their functionally dense, polar structures⁵¹. In contrast, the synthetic strategy for these non-nucleotide MSA-2 prodrugs described herein is straightforward and can

be readily produced on a gram scale from commercially available reagents. Hence, we can expect these compounds could be of significant interest for further clinical translation.

47. J. Guo, L. Huang, Nanodelivery of cGAS-STING activators for tumor immunotherapy. *Trends Pharmacol. Sci.* **43**, 957-972 (2022).

48. P. Zhang *et al.* STING agonist-loaded, CD47/PD-L1-targeting nanoparticles potentiate antitumor immunity and radiotherapy for glioblastoma. *Nat. Commun.* **14**, 1610 (2023).

49. Wehbe, M. *et al.* Nanoparticle delivery improves the pharmacokinetic properties of cyclic dinucleotide STING agonists to open a therapeutic window for intravenous administration. *J. Control. Release* **330**, 1118-1129 (2021).

50. Dane, E.L. *et al.* STING agonist delivery by tumour-penetrating PEG-lipid nanodiscs primes robust anticancer immunity. *Nat. Mater.* **21**, 710-720 (2022).

51. McIntosh, J.A. *et al.* A kinase-cGAS cascade to synthesize a therapeutic STING activator. *Nature* **603**, 439-444 (2022).

Minor comments:

1. Line 134: The authors present data on cytokine and chemokine expression from DCs in vitro, and then state “These data indicate that the SAProsome platform can facilitate tumor-infiltration and activation of CD8+ T-effector cells.” This is an overstatement, one can’t claim this from these in vitro cytokine/chemokine secretion data.

Response:

We greatly appreciate your valuable feedback, which helped to improve the quality of our manuscript. As per your suggestion, we have deleted this overstatement and revised the discussion as follows in the revised manuscript (page 7).

These experimental results indicate that intracellular esterase can spontaneously trigger the liberation of active MSA-2, which eventually facilitate the activation of STING pathway and lead to augmented secretion of cytokines.

2. Line 201: Text calls out Fig. 5I and J, this should be Fig. 4I and J.

Response:

Thanks for your careful reading. We have carefully revised the text throughout the manuscript and ‘Fig. 4I and J’ has been changed to ‘Fig. 4i and j’ in the revised manuscript.

Reply to comments of Reviewer #2 (Remarks to the Author):

Summary: The manuscript by Chen et al. describes a nanoparticle-based strategy for systemic (i.v.) delivery of STING agonist prodrugs based on the MSA-2 small molecule STING agonist. The authors use the carboxylic acid of MSA-2, which is critical for STING binding, to generate a small library of lipophilic ester-based prodrugs that are then encapsulated into a liposomal formulation for systemic delivery (referred to as "prosome"). Consistent with what is known about other prodrugs that must be cleaved by esterases to generate the active agent, they find that the rate of conversion from prodrug to MSA-2 is dependent on carbon spacer length. They next demonstrate that the prodrugs, which become highly lipophilic, can be loaded into liposomal nanoparticles and that enhance STING activation in cell culture models compared to the MSA-2 parent compound. The authors then test the efficacy of 4 prosome formulations injected intravenously in an MC38 tumor model, demonstrating enhanced activity relative to MSA-2 that is administered orally and also identify a lead formulation (SAProsome-3) based on its potent antitumor activity. The authors then progress to evaluating how the lead formulation impacts the tumor microenvironment, demonstrating enhanced infiltration of CD8+ T cells and NK cells as well as a shift towards an increased number of activated DCs and M1-like microphages. The authors also test the efficacy of the lead formulation in a more challenging tumor models of 4T1 breast cancer metastasis and B16F10 melanoma. Finally, the authors evaluate the PK and biodistribution, finding that the liposomal formulation extends half-life over the free MSA-2 and increases tumor and lymph node accumulation of the STING agonist.

Overall evaluation: Overall, this is a solid manuscript that follows a logical workflow for development and testing of a new immunotherapy formulation and is also supported by strong data in multiple mouse tumor models that demonstrates potent efficacy of the platform. Unfortunately, the manuscript suffers from a major flaw in that it compares the i.v. administered prodrug formulations only to free MSA-2 that is administered orally. This is not a fair comparison as it well-established that the bioavailability of most drugs delivered orally is lower than that delivered i.v. There should not be a technical reason for not comparing i.v. MSA-2 to the prodrug/prosome formulations since i.v. delivery was used in their PK studies and there are a number of ways in which hydrophobic agents can be solubilized for i.v. delivery (for example, use of PEG or other excipients). At minimum, a more appropriate comparison would have been to subQ administration of free MSA-2 at 50 mg/kg as was performed in the original Pan et al. paper where they observed robust responses using the subQ route. While there are several other major questions that need to be addressed that are outlined below, a significant

flaw of the current work is the lack of a comparison to free MSA-2 that is also delivered i.v., ideally at the MTD of the free agent but at minimum at the same dose. If such a comparison can be made and the data clearly show that the prodrug/liposomal formulations increases the efficacy or potency of MSA-2 then the manuscript may be suitable for publication provided other points are also addressed.

Response:

Many thanks for your critical reading and positive comments on our manuscript. Your valuable suggestions have greatly contributed to the improvement of the quality of this work. According to the reviewer's suggestions, we systemically compared the efficacy of free MSA-2 (intravenous injection or oral gavage) and SAProsome-3 (intravenous injection) at different doses. This efficacy testing was performed on an additional Lewis lung carcinoma (LLC) mouse model. These new data are included in **Fig. 6a-d** and **Supplementary Fig. S23** in the revised version. We found that in this model, intravenous administration of SAProsome-3 yielded durable tumor regression but intravenous free MSA-2 is less effective than SAProsome-3. We also confirmed that oral MSA-2 at 160 mg/kg, intravenous MSA-2 at 40 mg/kg, and intravenous SAProsome-3 at 30 mg/kg (MSA-2 equivalence) had the comparable antitumor efficacy in this animal model. However, it should be noted that free MSA-2 is not soluble in water, requiring DMSO as an injection solvent, which is not practical for clinical use. Please see these new results in the revised manuscript (Page 11-12).

Fig. 6. SAProsome-3 potentiates therapeutic efficacy and synergizes with checkpoint blockade inhibition. a–d LLC tumor-bearing mice were treated with free MSA-2 via intravenous (IV) administration or oral gavage, and SAProsome-3 following intravenous injection at indicated MSA-2–equivalent doses. Tumor growth curves **a**, spider plots of individual tumor growth curves **b**, Kaplan–Meier survival curves **c** and body weight changes **d** of tumor-bearing mice are shown (n = 6 mice/group).

Major Points:

1. Pertaining to above, the authors need to perform extensive additional experiments to compare the prosome formulations to free MSA-2 administered via i.v. route, or at minimum, via the subQ route used by Pan et al, which also demonstrated enhanced efficacy compared to oral administration. The authors perform i.v. administration in their PK/biodistribution study and so this delivery route is feasible. Without such direct comparison and a critical control, conclusions regarding the relative efficacy of prosome compared to the free active agent cannot be made and the impact of the technology cannot be assessed.

Response:

We greatly appreciate the reviewer for his/her valuable suggestions made on our

manuscript. We conducted comparative studies on the antitumor activity of MSA-2 administered intravenously or orally versus prosome formulation against a Lewis lung carcinoma (LLC) model. We have included a detailed description of our methodology and results in the revised manuscript (Page 11-12).

Efficacy comparison of SAProsome-3 versus free MSA-2 via different routes

We next sought to compare the therapeutic efficacy of SAProsome-3 *versus* free MSA-2. The mouse tumor model bearing Lewis lung carcinoma (LLC) was established and the animals were treated with different doses and administration routes of therapeutics. Dose-dependent antitumor activity was observed in each treatment group (**Fig. 6a** and **Supplementary Fig. S23**). Interestingly, in this model, we still observed the striking potency of intravenous SAProsome-3, yielding durable tumor regression and complete responses in 50% and 66.7% of mice at 35 and 40 mg/kg of MSA-2–equivalent doses, respectively (**Fig. 6a** and **b**). Notably, dosing of SAProsome-3 at 40 mg/kg led to significant tumor volume shrinkage to ~30 mm³. In sharp contrast, free MSA-2 administered either by oral or intravenous route only showed limited efficacy against LLC tumors (**Fig. 6a-c** and **Supplementary Fig. S23**). We also confirmed the doses of oral MSA-2 at 160 mg/kg, intravenous MSA-2 at 40 mg/kg, and intravenous SAProsome-3 at 30 mg/kg (MSA-2 equivalence), by which they had the comparable antitumor efficacy in this animal model (**Fig. 6a-c**). Histological analysis using H&E staining and TUNEL-positive cells labeling revealed pronounced intratumoral apoptosis after SAProsome-3 treatment (**Supplementary Fig. S24**). Moreover, administration of SAProsome-3 enhanced infiltration of CD8⁺ T cells in the tumors, which correlated positively with elevated production of granzyme B (**Supplementary Fig. S24**). Again, SAProsome-3 was proven to be safe for intravenous administration as supported by stable body weights in animals (**Fig. 6d**).

Fig. 6. SAProsome-3 potentiates therapeutic efficacy and synergizes with checkpoint blockade inhibition. a–d LLC tumor-bearing mice were treated with free MSA-2 via intravenous (IV) administration or oral gavage, and SAProsome-3 following intravenous injection at indicated MSA-2–equivalent doses. Tumor growth curves a, spider plots of

individual tumor growth curves **b**, Kaplan–Meier survival curves **c** and body weight changes **d** of tumor-bearing mice are shown (n = 6 mice/group). **e–h** Mice with ~150 mm³ subcutaneous B16F10 tumors were administered with a single dose of saline (IV), free MSA-2 (PO) or SAProsome3 (IV) equivalent to 35 mg/kg MSA-2. For combined treatment with the checkpoint blockade antibody, mice were additionally administered with anti-PD-L1 antibody intraperitoneally three times at a dose of 5 mg/kg (n = 6 mice/group). **f** Tumor-growth curves for individual mice. **g** Kaplan–Meier survival analysis (log-rank test). The mice were treated with the indicated formulations, and a tumor volume of 2000 mm³ was set as the endpoint criteria. **h** Body weight change of B16F10 tumor-bearing mice during treatment (n = 6 mice/group).

2. Another important control is MSA-2 loaded into liposomes; i.e., not a prodrug form. The authors claim that they were not able to load MSA-2 into liposomes and it is logical that converting into the prodrug format would increase lipophilicity and loading into liposomes. However, the pKa of the carboxylic acid of MSA-2 is 4.8 and therefore it will also be water insoluble at lower pH values and so presumably would load well into liposomes under these conditions. Have the authors attempted such alternatives for loading of MSA-2 into liposomes so that a comparison between a liposomal formulation and a prosome can be made? Such a study is important for determining the relative importance of the prodrug feature relative to the liposomal delivery system.

Response:

As the reviewer notes, including unmodified MSA-2 formulated in liposomes is an important control. In response to the reviewer's suggestion, we attempted to encapsulate free MSA-2 into liposomes under acidic conditions by adjusting the pH of the media, and the results are presented in the below Figure R1. We observed visible precipitates when free MSA-2 molecule at different pH values was attempted to be formulated into liposomes. In contrast, the solution containing SAProsome-3 at a concentration equivalent to 2 mg/mL MSA-2 remained stable before and after centrifugation. These results verified that unmodified MSA-2 is not miscible with liposomal compositions and the prodrug strategy is necessitated to alter the physicochemical property of MSA-2 for effective liposomal delivery.

Figure R1. A comparison was made between the water dispersion of MSA-2 and prodrug **3** when fabricated into liposomes. It was observed that at a concentration of 1 mg/ml and different pH values, MSA-2 liposomes produced precipitates in the solution, both before and after centrifugation (5000 rpm, 10 min). In contrast, the liposomal formulation of prodrug **3** (SAProsome-3) remained transparent even at a higher concentration of 2 mg/ml and pH7.4. S1: sample 1, S2: sample 2, S3: sample 3.

3. The authors claim that uncontrolled systemic inflammation is an issue for systemically administered STING agonists and that their design might mitigate this problem. However, the authors do not quantify levels of serum cytokines or the levels of STING-driven inflammation in major distribution organs and so the relative impact of the design on systemic inflammation compared to delivery of a free MSA-2 agonist is not known. These experiments need to be performed to help determine if the design can minimize systemic inflammatory effects.

Response:

According to this reviewer's comment, we quantified the levels of serum cytokines and STING-driven inflammation in major distribution organs to determine the potential of our design in minimizing systemic inflammatory effects compared to free MSA-2 agonist delivery. We confirmed the doses of oral MSA-2 at 160 mg/kg, intravenous MSA-2 at 40 mg/kg, and intravenous SAProsome-3 at 30 mg/kg (MSA-2 equivalence), which had comparable antitumor efficacy in the LLC model (**Fig. 6a-c**). Hence, a 1.5-fold higher dose than therapeutic dose was injected for evaluation of systemic inflammatory effects. In addition, toxicities to major organs using serum biochemical analysis and histological examination were also

assessed. The results and corresponding descriptions have been added to the revised manuscript (Page 14-15).

Nanoliposomal vesicles abrogate STING agonist-associated systemic toxicities

Finally, we investigated the potential toxicity induced by systemic administration of STING agonists. The same administration schedule with a 1.5-fold higher dose than therapeutically relevant doses used in the LLC model was tested. Serum biochemical analysis showed that following SAProsome-3 treatment, all parameters remained unchanged as compared to the healthy group (**Fig. 7f-g**). Conversely, the animals receiving intravenous MSA-2 experienced significant increases in aspartate aminotransferase (AST) and alanine aminotransferase (ALT), indicating obvious liver toxicities. Moreover, oral administration of MSA-2 also resulted in significantly elevated serum enzyme levels, including AST, ALT, urine acid (UA) and blood urea nitrogen (BUN). Histological examination of organ sections also suggested obvious toxicities to liver and kidney tissues after treatment with free MSA-2 *via* either intravenous or oral administration, while those treated with SAProsome-3 were indistinguishable from untreated animals (**Supplementary Fig. S26**). We found acute inflammation in free drug-treated groups, characterized by elevated levels of inflammatory cytokines/chemokines (**Fig. 7h** and **Supplementary Fig. S27**). Of note, IL-6 (a key driver of cytokine storm) maintained a 12.5 or 18.6-fold higher level in serum from mice treated with MSA-2 orally or intravenously compared to that of SAProsome-3 (**Fig. 7h**). In contrast, SAProsome-3 showed negligible fluctuation of these cytokines/chemokines in serum and healthy organs, indicating minimized systemic inflammation and immunotoxicities.

Figure 7. f-g C57BL/6 mice were treated with free MSA-2 orally at a dose of 240 mg/kg or intravenously at a dose of 60 mg/kg, and SAProsome-3 intravenously at a dose of 45 mg/kg per mouse, every three days for a total of three injections. Blood chemistry of mice were analyzed on day 8 postadministration. **h** The levels of IL-6 in the serum and other tissues were determined using ELISA at 6 h postadministration of free MSA-2 (PO, 240 mg/kg), free MSA-2 (IV, 60 mg/kg) or SAProsome-3 (IV, 45 mg/kg) (n = 5).

Meanwhile, the quantitative levels of TNF- α and IFN- γ in the serum and other tissues have been presented in the supplementary Figure S27.

Fig. S27. a-b The levels of TNF- α and IFN- γ in the serum and other tissues were determined using ELISA 6 h after a single administration of free MSA-2 (PO, 240 mg/kg), free MSA-2 (IV, 60 mg/kg) or SAProsome-3 (IV, 45 mg/kg) (n = 5 in each group).

4. Related to above, while the weight loss data provided suggests that the system is well-tolerated, additional analysis of toxicity, including serum markers of liver/kidney damage as well as organ pathology should be performed to clearly demonstrate relative toxicity of the system. Also, the body weight data (Fig 3C) should instead be plotted as % loss or gain from original mass.

Response:

Related to above response, we have assessed serum markers of liver/kidney damage, as well as organ pathology. While free MSA-2 demonstrated effective antitumor efficacy, a

higher dose was required than the prodrug liposome to elicit comparable activity, which may cause obvious dose-limiting side effects. In contrast, SAProsome-3 showed negligible *in vivo* toxicities such as liver and renal damages even when administered at 45 mg/kg (MSA-2 equivalence). Thus, the prosome formulation was expected to optimize the *in vivo* delivery of MSA-2 agonist against cancer, triggering substantial antitumor immune responses while without inducing systemic toxicities. This prosome platform could be beneficial for further clinical translation. We have included these results and relevant descriptions in the revised manuscript (Page 14-15).

Nanoliposomal vesicles abrogate STING agonist-associated systemic toxicities

Finally, we investigated the potential toxicity induced by systemic administration of STING agonists. The same administration schedule with a 1.5-fold higher dose than therapeutically relevant doses used in the LLC model was tested. Serum biochemical analysis showed that following SAProsome-3 treatment, all parameters remained unchanged as compared to the healthy group (**Fig. 7f-g**). Conversely, the animals receiving intravenous MSA-2 experienced significant increases in aspartate aminotransferase (AST) and alanine aminotransferase (ALT), indicating obvious liver toxicities. Moreover, oral administration of MSA-2 also resulted in significantly elevated serum enzyme levels, including AST, ALT, urine acid (UA) and blood urea nitrogen (BUN). Histological examination of organ sections also suggested obvious toxicities to liver and kidney tissues after treatment with free MSA-2 *via* either intravenous or oral administration, while those treated with SAProsome-3 were indistinguishable from untreated animals (**Supplementary Fig. S26**). We found acute inflammation in free drug-treated groups, characterized by elevated levels of inflammatory cytokines/chemokines (**Fig. 7h** and **Supplementary Fig. S27**). Of note, IL-6 (a key driver of cytokine storm) maintained a 12.5 or 18.6-fold higher level in serum from mice treated with MSA-2 orally or intravenously compared to that of SAProsome-3 (**Fig. 7h**). In contrast, SAProsome-3 showed negligible fluctuation of these cytokines/chemokines in serum and healthy organs, indicating minimized systemic inflammation and immunotoxicities.

Figure 7. f-g C57BL/6 mice were treated with free MSA-2 orally at a dose of 240 mg/kg or intravenously at a dose of 60 mg/kg, and SAProsome-3 intravenously at a dose of 45 mg/kg per mouse, every three days for a total of three injections. Blood chemistry of mice were analyzed on day 8 postadministration. h The levels of IL-6 in the serum and other tissues were determined using ELISA at 6 h postadministration of free MSA-2 (PO, 240 mg/kg), free MSA-2 (IV, 60 mg/kg) or SAProsome-3 (IV, 45 mg/kg) (n = 5).

Fig. S26. Representative images of H&E staining of major organs (heart, lung, liver, and kidney) excised from C57BL/6 mice after different treatments.

Additionally, we have represented the change in body weight more accurately by plotting the body weight data as % loss in the revised manuscript (**Figure 3c**).

5. Considering that the rest of the studies, including biodistribution studies, were performed in mice, it was not clear why the pharmacokinetic analysis was done in rats. It would have been much more instructive to do this in mice so that relationships between pharmacological properties, efficacy, and toxicity can be drawn.

Response:

Many thanks for your suggestions. Using rats for the pharmacokinetic analysis was based on the requirement of sufficient blood samples for subsequent HPLC analysis. Collecting approximately 500 μ L of blood sample in mice needs to sacrifice one mouse at each time point, which is unfavorable for ethical and animal welfare reasons. Therefore, to reduce the number of experimental animals, we have previously used rats that have sufficient blood volume for analysis to measure drug plasma concentrations (Please refer to our work: Proc Natl Acad Sci U S A 120, e2210385120 (2023); Adv Sci, e2204890 (2023); Biomaterials 290, 121814 (2022); J Control Release 328, 237-250 (2020); ACS Nano 16, 10242-10259 (2022); Am J Transplant 21, 3871-3882 (2021); etc.). In addition, there are examples that used rats for pharmacokinetic analysis (e.g., Nature Communications, 2023, 14.1: 255; Nature communications, 2021, 12.1: 3453; Nature Biomedical Engineering, 2019, 3.4: 264-280; Nature biomedical engineering, 2021, 5.3: 252-263; Science advances, 2020, 6.45: eabc1725; ACS nano, 2022, 16.12: 21417-21430; Proc Natl Acad Sci U S A 2011, 108.5: 1850-1855; etc.).

We also agree with the reviewer that further experiments and evaluations are necessary

to comprehensively understand the relationships between pharmacological properties, efficacy, and toxicity. The suggestion to use the same species for both pharmacokinetic and biodistribution studies is highly appreciated.

6. Related to above, the authors also need to evaluate the biodistribution of drug in other distribution organs such as liver, spleen, and potentially lung; only tumor and LNs are shown and, while these are important, the ratio of drug in these tissues relative to that in other organs is more important. Further, in the original Pan et al. paper it is shown that free MSA-2 preferentially accumulates in tumor tissue due to change in solubility due to protonation of the carboxy group within the tumor microenvironment. So, an evaluation of drug distribution to other organs and the ratio of drug in tumor/other tissues needs to be performed.

Response:

Thanks for this valuable suggestion. We investigated the biodistribution of the drug in other organs, including the heart, liver, spleen, lung, and kidney in the MC38 model at 8 hours post-administration. Our results demonstrated that SAProsome-3 exhibited a higher accumulation in the tumor and LNs, while showing lower distribution in the heart and kidney, in comparison to free MSA-2. Remarkably, in the tumor of mice treated with SAProsome-3, MSA-2 and its intact form maintained a 75.3 and 73.8-fold higher concentration compared to that in the heart and kidney, respectively (**Fig. 7c** and **Supplementary Fig. S25**). In contrast, the concentration of MSA-2 in the tumor was lower than the concentration in the heart and kidney of mice treated intravenously with free MSA-2.

Also, there are no quantitative data of drug concentration in other tissues except for the tumor and plasma in the original Pan et al. paper. We found that the MSA-2 concentration in the tumor at 4 h after oral administration at a dose of 60 mg/kg was approximately 0.2 μM (i.e., 59 ng/g), which was comparable to our data of the MSA-2 concentration (i.e., 10 ng/g) at a dose of 17.5 mg/kg at 8 h after oral administration, indicating the reproducibility and reliability between these data.

In addition, we have calculated the ratios of drug concentrations in tumors *versus* other organs for SAProsome-3 and free MSA-2 (administered either intravenously or orally). The data revealed significant increases in the tumor/other tissues ratios in the SAProsome-3 group compared to both free MSA-2 groups, manifesting the preferential accumulation of SAProsome-3 in target tumor lesions and unfavorable tissue biodistribution of free drugs.

Therefore, these data suggest that the prodrug-formulated liposome (i.e., SAProsome-3) has a favorable distribution profile that can potentially increase intratumoral drug accumulation while minimizing drug exposure in off-target organs. We have included these results and analysis in the revised manuscript on Page 14, as follows.

On the other hand, SAProsome-3 had low distribution across other normal organs, whereas intravenously administered MSA-2 accumulated at a higher concentration in the heart and kidneys (**Supplementary Fig. S25**).

Figure S25. Mice bearing MC38 xenograft tumors were given a single dose of MSA-2 or SAProsome-3 at a 17.5 mg/kg of MSA-2 equivalent dose. Drug concentration in major organs was determined using HPLC analysis at 8 h after administration. Ratios of drug concentrations in tumors *versus* other tissues (e.g., heart, liver, spleen, lung and kidney) are shown in purple on the bars. Data represent the means \pm SD (n = 6).

7. In Figure 2C, the activity of prosomes vs. free MSA2 are shown at 2 different concentrations. First, this (or similar) should be done over a wider range of concentrations and an EC₅₀ estimated for the different groups; this will allow for the relative potencies of each to be determined, which is an important metric when comparing new formulations. Second, statistical analysis of this data is missing.

Response:

We appreciate the reviewer's constructive comments and have conducted additional experiments to determine the EC₅₀ values and relative potency of SAProsomes and free MSA-2. We used various concentrations of MSA-2 and SAProsomes to measure IFN- β secretion in

cell culture and calculated the EC₅₀ values using the GraphPad Prism software.

We also conducted statistical analysis of the data in **Fig. 2c** using a one-way ANOVA analysis, and the results showed significant differences between SAProsomes and free MSA-2 at both tested concentrations in terms of *IFNβ1* mRNA expression. These new data were incorporated into **Fig. 2d** and relevant description was presented in the revised manuscript (Page 6) as follows.

In addition, dose-dependent secretion of the critical cytokine, IFN-β, after treatment with SAProsomes was observed in BMDCs (Fig. 2d), which allowed to identify the superior activity of SAProsome-3 and 4 with relatively low half maximal effective concentration (EC₅₀).

Figure 2. c STING-dependent downstream expression of inflammatory cytokines in BMDCs treated with MSA-2 or SAProsomes as assessed via qRT-PCR. **d** Dose–response curves of IFN-β secretion upon various treatments. The concentration of IFN-β in BMDC culture medium was determined via ELISA.

8. In Figure 2I, a control should be run using either non-Ova-pulsed DCs and/or B16 cells that do not express Ova in order to demonstrate that the killing response is antigen specific. It is somewhat surprising that there would be a sufficient number of Ova-specific T cells in a wild-type mouse that can be expanded in vitro to kill target cells. Therefore, it is possible that what is being observed is non-specific target cell killing; the controls mentioned above are necessary to demonstrate this.

Response:

We agree with the reviewer’s comment to include additional controls to validate the antigen-specific killing response presented in Figure 2I. To address this concern, we included B16F10 cells that do not express OVA as a control. Our results demonstrated a significant decrease in the killing effect on these control cells relative to the treatment of B16F10-OVA,

suggesting that the enhanced killing response is mediated *via* the OVA antigen. We have supplemented these results along with the relevant discussion in the revised manuscript (Page 7) as follows.

Crucially, pretreatment of BMDCs with SAProsomes and a tumor-specific antigen (e.g., OVA) significantly enhanced the cytotoxicity of splenic T-cells against tumor cells (B16F10-OVA), as compared to untreated cells (**Fig. 2h**). In contrast, splenic T-cells had limited cytotoxicity against B16F10 tumor cells. These experimental results indicate that intracellular esterase can spontaneously trigger the liberation of active MSA-2, which eventually facilitate the activation of STING pathway and lead to an augmented expression of cytokines.

Figure 2. h Cotreatment with the OVA peptide and SAProsomes increased the B16F10-OVA cell lysis via cytotoxic T lymphocytes *in vitro*. Experimental protocol is shown on left-hand side. Cell cytotoxicity was determined based on the lactate dehydrogenase (LDH) concentration in medium supernatant, as measured by the LDH assay. The cell number ratios (T-cell/B16F10 or B16F10-OVA cell/BMDC) were fixed at 10:1:5. Statistical significance was analyzed using one-way analysis of variance.

9. A technical consideration is why the percentage of CD86+CD80+ DCs is so high (~25%) in the untreated group (Figure 2G); this may suggest a need to better titrate antibodies or perhaps another source of innate immune activation (e.g., cell stress, contaminants in cell culture).

Response:

We have carefully examined our experimental protocols and found no indication of contamination or cell stress. However, to address the concern raised by the reviewer, we optimized our incubation time of BMDCs from 8 days to 6 days to reduce the innate immune activation. We performed additional experiments to revalidate the percentage of CD86+CD80+ DCs, and the results showed decreased levels of CD86+CD80+ DCs in the

untreated group. We appreciate the reviewer's comment, and we have updated the results in the revised manuscript.

Figure 2. f Following treatment with SAProsomes, BMDCs exhibit significantly increased activation markers, including major histocompatibility complex II (MHC II) **e** and CD80/CD86 **f** (n = 3).

10. A challenge with ester-based prodrugs is potential to be cleaved by murine esterases but not human esterases. The authors should confirm prodrug activity in human cells as well.

Response:

Thanks for the comment. As the reviewer notes, validation of ester-based prodrugs by human esterases is necessary. We therefore conducted experiments in the human monocytic cell line THP1 to assess the cleavability and activity of prodrug **3**. The following experiments were used to compare the STING-activating activity of free MSA-2 and SAProsome-3. The *in vitro* results indicate that prodrug **3** can be spontaneously cleaved to release active MSA-2 in response to human esterase and to trigger the activation of STING pathway. We have incorporated these findings and descriptions in the revised manuscript (Page 6-7).

A human monocytic cell line THP1 was further utilized to validate the potency of SAProsomes in activating STING. The results found that SAProsome-3 exhibited the fast intracellular breakdown of prodrug **3** and release of MSA-2 as shown in **Supplementary Fig. S12**. Moreover, SAProsome-3 was able to upregulate the mRNA expression of IFN- β , TNF α and CXCL10 that are regulated by STING, as well as activate proteins associated with the STING pathway in THP1. Notably, SAProsomes also significantly enhanced IFN- β secretion, which indicated their potential as the potent STING activator (**Supplementary Fig. S13**).

Fig. S12. **a** HPLC analysis was used to measure drug activation in THP1 cells. The cells were incubated with SAProsome-3 for either 1 or 2 hours. **b** Percentages of activated MSA-2 in THP1 cells were determined using their corresponding standard curves. The data are presented as the mean \pm SD ($n = 3$).

Fig. S13. **a–b** Human THP1 cells were incubated with either free MSA-2 or SAProsome-3 for 6 hours. The activation of the STING pathway was further analyzed using western blot analysis **a** and quantitative real-time PCR analysis **b** of gene expression. **c** IFN- β levels in cell culture media of human THP1 treated with the indicated concentrations of MSA-2 formulations were determined by ELISA.

11. A more minor point is that the authors are encouraged to expand upon their discussion as to why prosome3 seems to have the best therapeutic efficacy of the 4 tested. This is an interesting finding and so the authors should offer some reasons as to why this might be the case.

Response:

We sincerely thank the reviewer for this comment. We have included relevant discussions in the revised manuscript (Page 17-18) as follows.

We disclosed that the overall chemical structures of prodrugs affected the hydrolysis kinetics and activation of MSA-2 in response to esterase. In fact, a positive correlation between hydrolysis rate and STING-stimulating activity was observed in this study. Moreover, SAProsome-3 had the highest AUC_{0-t} , which could, in part, explain its superior antitumor efficacy. Despite these findings, the structure-activity relationships remain to be explored in depth.

REVIEWERS' COMMENTS

Reviewer #1 (Remarks to the Author):

The authors have done an admirable job responding to the review critiques with additional data and discussion. I feel the paper is substantially approved and recommend acceptance.

Reviewer #2 (Remarks to the Author):

The authors have adequately addressed all of my comments and have included a number of new experiments that have strengthened the conclusions and rigor of the work. Overall, this work appears to represent an important advance in enhancing the efficacy of systemically administered STING agonists.

Reply to comments of Reviewer #1 (Remarks to the Author):

This manuscript by Chen et al. reports on the development of a strategy for systemic delivery of a non-nucleotide small molecule STING agonist, MSA-2, via encapsulation of an MSA-2-prodrug in liposomal vehicles. The authors test different liposomal compositions and identify one, Saprosome-3, which is shown to be safe and effective in concentrating MSA-2 in tumors, leading to impressive anti-tumor activity in models of primary tumor treatment and prevention of metastatic recurrence, accompanied by induction of protective memory. Overall this strategy appears promising and the prodrug+liposome formulation is elegant.

I do have one major issue with the data as presented—the comparison to per-oral dosing. The original publication describing MSA-2 reported that it was safe when dosed to high levels orally or subcutaneously, but oral dosing at 20 mg/kg 3X was completely ineffective in tumor control while oral dosing at 80 mg/kg 3X elicited strong anti-tumor activity. The oral dosing used here is not completely clear- the liposomes are dosed at 35 mg/kg (MSA-2 equivalent dose), and it seems the authors may be using the same dose for oral free MSA-2. But this is not the comparison that should be made, as 35 mg/kg may be too low for efficacy when given orally—the liposomes at their MTD should be compared to free oral MSA-2 at its previously reported effective dose (i.e., order of 80 mg/kg). For clinical translation, it would be more attractive to give free MSA-2 at a higher dose orally than having to resort to i.v. administration, if the oral route gives the same tumor response without toxicity. There is no obvious benefit to the liposomes if their main impact is to allow a lower dose to be effective, if there aren't dose-limiting side effects of oral dosing at a higher dose.

Response:

We would like to express our sincere appreciation to the reviewer for taking the time to review our manuscript and for providing positive feedback on our work. We also appreciate the reviewer for these useful comments made on our manuscript. In the original submission, we focused the studies on the efficacy testing at the same doses of MSA-2 in different tumor models, while having ignored the toxicity evaluation. We agree with the reviewer the opinion that it would be more attractive to give free MSA-2 at a higher dose orally. We therefore compared the antitumor activity of different administration routes to at different doses and sought to determine the doses of each formulation that have equal antitumor efficacy.

For this purpose, we conducted the efficacy studies against a Lewis lung carcinoma (LLC) model. We observed the dose-dependent antitumor activity for each MSA-2 formulation and

administration route. Interestingly, low doses of SAProsome-3 such as 35 or 40 mg/kg potently suppressed the tumor growth in this LLC model. These results and relevant description are supplemented in the revised manuscript (Page 11-12) and **supplementary Figures S23 and S24** as follows.

Efficacy comparison of SAProsome-3 versus free MSA-2 via different routes

We next sought to compare the therapeutic efficacy of SAProsome-3 *versus* free MSA-2. The mouse tumor model bearing Lewis lung carcinoma (LLC) was established and the animals were treated with different doses and administration routes of therapeutics. Dose-dependent antitumor activity was observed in each treatment group (**Fig. 6a** and **Supplementary Fig. S23**). Interestingly, in this model, we still observed the striking potency of intravenous SAProsome-3, yielding durable tumor regression and complete responses in 50% and 66.7% of mice at 35 and 40 mg/kg of MSA-2–equivalent doses, respectively (**Fig. 6a** and **b**). Notably, dosing of SAProsome-3 at 40 mg/kg led to significant tumor volume shrinkage to ~30 mm³. In sharp contrast, free MSA-2 administered either by oral or intravenous route only showed limited efficacy against LLC tumors (**Fig. 6a-c** and **Supplementary Fig. S23**). We also confirmed the doses of oral MSA-2 at 160 mg/kg, intravenous MSA-2 at 40 mg/kg, and intravenous SAProsome-3 at 30 mg/kg (MSA-2 equivalence), in which they had the comparable antitumor efficacy in this animal model (**Fig. 6a-c**). Histological analysis using H&E staining and TUNEL-positive cells labeling revealed pronounced intratumoral apoptosis after SAProsome-3 treatment (**Supplementary Fig. S24**). Moreover, administration of SAProsome-3 enhanced infiltration of CD8⁺ T cells in the tumors, which correlated positively with elevated production of granzyme B (**Supplementary Fig. S24**). Again, SAProsome-3 was proven to be safe for intravenous administration as supported by stable body weights in animals (**Fig. 6d**).

Fig. 6. SAProsome-3 potentiates therapeutic efficacy and synergizes with checkpoint blockade inhibition. a–d LLC tumor-bearing mice were treated with free MSA-2 via intravenous (IV) administration or oral gavage, and SAProsome-3 following intravenous injection at indicated MSA-2–equivalent doses. Tumor growth curves **a**, spider plots of individual tumor growth curves **b**, Kaplan–Meier survival curves **c** and body weight changes

d of tumor-bearing mice are shown (n = 6 mice/group). **e–h** Mice with ~150 mm³ subcutaneous B16F10 tumors were administered with a single dose of saline (IV), free MSA-2 (PO) or SAProsome3 (IV) equivalent to 35 mg/kg MSA-2. For combined treatment with the checkpoint blockade antibody, mice were additionally administered with anti-PD-L1 antibody intraperitoneally three times at a dose of 5 mg/kg (n = 6 mice/group). **f** Tumor-growth curves for individual mice. **g** Kaplan–Meier survival analysis (log-rank test). The mice were treated with the indicated formulations, and a tumor volume of 2000 mm³ was set as the endpoint criteria. **h** Body weight change of B16F10 tumor-bearing mice during treatment (n = 6 mice/group).

We also agree with the reviewer that it would be more appealing to administer free MSA-2 at a higher dose orally rather than having to resort to intravenous administration, if the oral route gives the same tumor response without toxicity. According to this comment, we investigated the potential toxicity induced by oral or intravenous administration of free MSA-2, which was compared with intravenously administered SAProsome-3. In the LLC model, we have confirmed that oral MSA-2 at 160 mg/kg, intravenous MSA-2 at 40 mg/kg, and intravenous SAProsome-3 at 30 mg/kg (MSA-2 equivalence) showed comparable antitumor activity (**Fig. 6a-c**). Hence, we assessed the toxicity of drugs at 1.5-fold higher doses than the abovementioned therapeutically relevant doses.

As the reviewer notes, although free MSA-2 taken orally was able to induce antitumor activity, a higher dose was required, which may cause dose-limiting side effects in animals. The results of *in vivo* toxicity studies showed that free drug treatment induced acute inflammation, characterized by elevated levels of inflammatory cytokines/chemokines in the blood and major organs. In addition, hepatic and renal toxicities were also observed in free MSA-2–treated mice. In sharp contrast, SAProsome-3 had negligible *in vivo* toxicities such as liver and renal damages and system inflammation when administered at 45 mg/kg (MSA-2 equivalence). Therefore, we expected that the SAProsome scaffold could optimize the *in vivo* delivery of MSA-2 agonist against cancer, triggering substantial antitumor immune responses within the safe dosing range. We have included these results in the revised manuscript (Page 14-15) shown as below.

Nanoliposomal vesicles abrogate STING agonist-associated systemic toxicities

Finally, we investigated the potential toxicity induced by systemic administration of STING agonists. The same administration schedule with a 1.5-fold higher dose than therapeutically

relevant doses used in the LLC model was tested. Serum biochemical analysis showed that following SAProsome-3 treatment, all parameters remained unchanged as compared to the healthy group (**Fig. 7f-g**). Conversely, the animals receiving intravenous MSA-2 experienced significant increases in aspartate aminotransferase (AST) and alanine aminotransferase (ALT), indicating obvious liver toxicities. Moreover, oral administration of MSA-2 also resulted in significantly elevated serum enzyme levels, including AST, ALT, urine acid (UA) and blood urea nitrogen (BUN). Histological examination of organ sections also suggested obvious toxicities to liver and kidney tissues after treatment with free MSA-2 via either intravenous or oral administration, while those treated with SAProsome-3 were indistinguishable from untreated animals (**Supplementary Fig. S26**). We found acute inflammation in free drug-treated groups, characterized by elevated levels of inflammatory cytokines/chemokines (**Fig. 7h** and **Supplementary Fig. S27**). Of note, IL-6 (a key driver of cytokine storm) maintained a 12.5 or 18.6-fold higher level in serum from mice treated with MSA-2 orally or intravenously compared to that of SAProsome-3 (**Fig. 7h**). In contrast, SAProsome-3 showed negligible fluctuation of these cytokines/chemokines in serum and healthy organs, indicating minimized systemic inflammation and immunotoxicities.

Fig. 7. Pharmacokinetic properties, biodistribution and safety profiles of SAProsomes.
a Drug concentration in the plasma of Sprague-Dawley rats ($n = 3$ rats/group) following a single administration of different drugs at a 17.5 mg/kg of MSA-2 equivalent dose. **b** Pharmacokinetic parameters reflecting half-life ($t_{1/2}$), C_0 , C_{max} , area under the curve (AUC_{0-t}) and mean residence time (MRT). Mice bearing MC38 **c**, 4T1 **d**, or B16F10 **e** xenograft tumors

were given a single dose of MSA-2 and SAProsome-3. Drug concentration in tumors and lymph nodes was determined using HPLC analysis at 8 and 24 h after administration. **f-g** C57BL/6 mice were treated with free MSA-2 orally at a dose of 240 mg/kg or intravenously at a dose of 60 mg/kg, and SAProsome-3 intravenously at a dose of 45 mg/kg per mouse, every three days for a total of three injections. Blood chemistry of mice were analyzed on day 8 postadministration. **h** The levels of IL-6 in the serum and other tissues were determined using ELISA at 6 h postadministration of free MSA-2 (PO, 240 mg/kg), free MSA-2 (IV, 60 mg/kg) or SAProsome-3 (IV, 45 mg/kg) (n = 5).

Other main issues:

1. Line 157, the authors write: “SAProsome treatment engendered complete tumor regression (CR), and the tumor inhibition rates were highly dependent on the encapsulated MSA-2 prodrugs.” Control treatment with empty liposomes is needed to confirm that the liposome doses used to achieve these high drug doses (35 mg/kg) are not directly having an anti-tumor effect.

Response:

Thanks for this comment. We accordingly evaluated the antitumor effect of empty liposomes in the MC38 model. We have incorporated this result in the revised manuscript (Page 8) and included the data in **supplementary Figure S14**.

As a comparison, empty liposomes without prodrug payloads did not show any antitumor benefits in this model (**Supplementary Fig. S14**).

Fig. S14. Tumor growth curves of MC38 xenografts in mice. Mice were treated with either saline or empty liposomes without prodrug payloads.

2. There is substantial recent prior work on formulating STING agonists of different types for enhanced delivery to tumors, and the authors relegate this work to a single sentence citing one review on the field (ref 47). The authors largely dismiss this other work, indicating that "...as CDN-based small-molecule agonists are hydrophilic and negatively charged, the therapeutic delivery of these polar agents remains technically challenging." Yet here, MSA-2 also required synthesis of a prodrug and formulation in liposomes for effective delivery, which suggests this compound's delivery is equally technically challenging. A deeper discussion of the pros and cons of the current approach vs. other publications is needed.

Response:

We acknowledge that there is substantial prior work on formulation of STING agonists for better delivery to tumors. According to this comment, we have added more discussion on the advantages and limitations of our approach compared to other publications. We agree with the reviewer that our approach also requires the synthesis of a prodrug and its formulation in liposomes for effective delivery. However, compared with CDN-based STING agonists, the synthesis and preparation protocol for the MSA-2 prodrug and its liposomes is straightforward and simple. The MSA-2 prodrugs can be readily produced from commercially available reagents using a two-step reaction protocol, which is beneficial for future clinical translation. On the other hand, liposomes are the most promising drug delivery carriers for clinical use. We here found that the rationally engineered STING prodrugs can be stably incorporated into liposomal vesicles, enabling intravenous injection for *in vivo* studies. Compared to the immiscibility of free MSA-2 with lipid constituents, our approach enables better pharmacokinetic properties and efficient delivery of MSA-2 to tumors or lymph nodes in multiple tumor models. Hence, we can expect these compounds to exhibit a high potential for further clinical translation. We appreciate the reviewer's valuable feedback and have incorporated the relevant discussion into the revised manuscript (Page 15-16) as follows.

Recently, substantial effort has been devoted to the discovery of STING agonists and various sophisticated delivery vehicles as antitumor therapeutics⁴⁷⁻⁵⁰. CDN-based agonists are hydrophilic and negatively charged, nanoparticle formulation and systemic delivery of these polar agents for therapeutic purposes remain challenging. In addition, the scalable synthesis of cGAMP analogues has posed substantial obstacles, requiring a tedious and lengthy synthetic protocol due to their functionally dense, polar structures⁵¹. In contrast, the synthetic strategy for these non-nucleotide MSA-2 prodrugs described herein is straightforward and can

be readily produced on a gram scale from commercially available reagents. Hence, we can expect these compounds could be of significant interest for further clinical translation.

47. J. Guo, L. Huang, Nanodelivery of cGAS-STING activators for tumor immunotherapy. *Trends Pharmacol. Sci.* **43**, 957-972 (2022).

48. P. Zhang *et al.* STING agonist-loaded, CD47/PD-L1-targeting nanoparticles potentiate antitumor immunity and radiotherapy for glioblastoma. *Nat. Commun.* **14**, 1610 (2023).

49. Wehbe, M. *et al.* Nanoparticle delivery improves the pharmacokinetic properties of cyclic dinucleotide STING agonists to open a therapeutic window for intravenous administration. *J. Control. Release* **330**, 1118-1129 (2021).

50. Dane, E.L. *et al.* STING agonist delivery by tumour-penetrating PEG-lipid nanodiscs primes robust anticancer immunity. *Nat. Mater.* **21**, 710-720 (2022).

51. McIntosh, J.A. *et al.* A kinase-cGAS cascade to synthesize a therapeutic STING activator. *Nature* **603**, 439-444 (2022).

Minor comments:

1. Line 134: The authors present data on cytokine and chemokine expression from DCs in vitro, and then state “These data indicate that the SAProsome platform can facilitate tumor-infiltration and activation of CD8+ T-effector cells.” This is an overstatement, one can’t claim this from these in vitro cytokine/chemokine secretion data.

Response:

We greatly appreciate your valuable feedback, which helped to improve the quality of our manuscript. As per your suggestion, we have deleted this overstatement and revised the discussion as follows in the revised manuscript (page 7).

These experimental results indicate that intracellular esterase can spontaneously trigger the liberation of active MSA-2, which eventually facilitate the activation of STING pathway and lead to augmented secretion of cytokines.

2. Line 201: Text calls out Fig. 5I and J, this should be Fig. 4I and J.

Response:

Thanks for your careful reading. We have carefully revised the text throughout the manuscript and ‘Fig. 4I and J’ has been changed to ‘Fig. 4i and j’ in the revised manuscript.

Reply to comments of Reviewer #2 (Remarks to the Author):

Summary: The manuscript by Chen et al. describes a nanoparticle-based strategy for systemic (i.v.) delivery of STING agonist prodrugs based on the MSA-2 small molecule STING agonist. The authors use the carboxylic acid of MSA-2, which is critical for STING binding, to generate a small library of lipophilic ester-based prodrugs that are then encapsulated into a liposomal formulation for systemic delivery (referred to as "prosome"). Consistent with what is known about other prodrugs that must be cleaved by esterases to generate the active agent, they find that the rate of conversion from prodrug to MSA-2 is dependent on carbon spacer length. They next demonstrate that the prodrugs, which become highly lipophilic, can be loaded into liposomal nanoparticles and that enhance STING activation in cell culture models compared to the MSA-2 parent compound. The authors then test the efficacy of 4 prosome formulations injected intravenously in an MC38 tumor model, demonstrating enhanced activity relative to MSA-2 that is administered orally and also identify a lead formulation (SAProsome-3) based on its potent antitumor activity. The authors then progress to evaluating how the lead formulation impacts the tumor microenvironment, demonstrating enhanced infiltration of CD8+ T cells and NK cells as well as a shift towards an increased number of activated DCs and M1-like microphages. The authors also test the efficacy of the lead formulation in a more challenging tumor models of 4T1 breast cancer metastasis and B16F10 melanoma. Finally, the authors evaluate the PK and biodistribution, finding that the liposomal formulation extends half-life over the free MSA-2 and increases tumor and lymph node accumulation of the STING agonist.

Overall evaluation: Overall, this is a solid manuscript that follows a logical workflow for development and testing of a new immunotherapy formulation and is also supported by strong data in multiple mouse tumor models that demonstrates potent efficacy of the platform. Unfortunately, the manuscript suffers from a major flaw in that it compares the i.v. administered prodrug formulations only to free MSA-2 that is administered orally. This is not a fair comparison as it well-established that the bioavailability of most drugs delivered orally is lower than that delivered i.v. There should not be a technical reason for not comparing i.v. MSA-2 to the prodrug/prosome formulations since i.v. delivery was used in their PK studies and there are a number of ways in which hydrophobic agents can be solubilized for i.v. delivery (for example, use of PEG or other excipients). At minimum, a more appropriate comparison would have been to subQ administration of free MSA-2 at 50 mg/kg as was performed in the original Pan et al. paper where they observed robust responses using the subQ route. While there are several other major questions that need to be addressed that are outlined below, a significant

flaw of the current work is the lack of a comparison to free MSA-2 that is also delivered i.v., ideally at the MTD of the free agent but at minimum at the same dose. If such a comparison can be made and the data clearly show that the prodrug/liposomal formulations increases the efficacy or potency of MSA-2 then the manuscript may be suitable for publication provided other points are also addressed.

Response:

Many thanks for your critical reading and positive comments on our manuscript. Your valuable suggestions have greatly contributed to the improvement of the quality of this work. According to the reviewer's suggestions, we systemically compared the efficacy of free MSA-2 (intravenous injection or oral gavage) and SAProsome-3 (intravenous injection) at different doses. This efficacy testing was performed on an additional Lewis lung carcinoma (LLC) mouse model. These new data are included in **Fig. 6a-d** and **Supplementary Fig. S23** in the revised version. We found that in this model, intravenous administration of SAProsome-3 yielded durable tumor regression but intravenous free MSA-2 is less effective than SAProsome-3. We also confirmed that oral MSA-2 at 160 mg/kg, intravenous MSA-2 at 40 mg/kg, and intravenous SAProsome-3 at 30 mg/kg (MSA-2 equivalence) had the comparable antitumor efficacy in this animal model. However, it should be noted that free MSA-2 is not soluble in water, requiring DMSO as an injection solvent, which is not practical for clinical use. Please see these new results in the revised manuscript (Page 11-12).

Fig. 6. SAProsome-3 potentiates therapeutic efficacy and synergizes with checkpoint blockade inhibition. a–d LLC tumor-bearing mice were treated with free MSA-2 via intravenous (IV) administration or oral gavage, and SAProsome-3 following intravenous injection at indicated MSA-2–equivalent doses. Tumor growth curves **a**, spider plots of individual tumor growth curves **b**, Kaplan–Meier survival curves **c** and body weight changes **d** of tumor-bearing mice are shown (n = 6 mice/group).

Major Points:

1. Pertaining to above, the authors need to perform extensive additional experiments to compare the prosome formulations to free MSA-2 administered via i.v. route, or at minimum, via the subQ route used by Pan et al, which also demonstrated enhanced efficacy compared to oral administration. The authors perform i.v. administration in their PK/biodistribution study and so this delivery route is feasible. Without such direct comparison and a critical control, conclusions regarding the relative efficacy of prosome compared to the free active agent cannot be made and the impact of the technology cannot be assessed.

Response:

We greatly appreciate the reviewer for his/her valuable suggestions made on our

manuscript. We conducted comparative studies on the antitumor activity of MSA-2 administered intravenously or orally versus prosome formulation against a Lewis lung carcinoma (LLC) model. We have included a detailed description of our methodology and results in the revised manuscript (Page 11-12).

Efficacy comparison of SAProsome-3 versus free MSA-2 via different routes

We next sought to compare the therapeutic efficacy of SAProsome-3 *versus* free MSA-2. The mouse tumor model bearing Lewis lung carcinoma (LLC) was established and the animals were treated with different doses and administration routes of therapeutics. Dose-dependent antitumor activity was observed in each treatment group (**Fig. 6a** and **Supplementary Fig. S23**). Interestingly, in this model, we still observed the striking potency of intravenous SAProsome-3, yielding durable tumor regression and complete responses in 50% and 66.7% of mice at 35 and 40 mg/kg of MSA-2–equivalent doses, respectively (**Fig. 6a** and **b**). Notably, dosing of SAProsome-3 at 40 mg/kg led to significant tumor volume shrinkage to ~30 mm³. In sharp contrast, free MSA-2 administered either by oral or intravenous route only showed limited efficacy against LLC tumors (**Fig. 6a-c** and **Supplementary Fig. S23**). We also confirmed the doses of oral MSA-2 at 160 mg/kg, intravenous MSA-2 at 40 mg/kg, and intravenous SAProsome-3 at 30 mg/kg (MSA-2 equivalence), by which they had the comparable antitumor efficacy in this animal model (**Fig. 6a-c**). Histological analysis using H&E staining and TUNEL-positive cells labeling revealed pronounced intratumoral apoptosis after SAProsome-3 treatment (**Supplementary Fig. S24**). Moreover, administration of SAProsome-3 enhanced infiltration of CD8⁺ T cells in the tumors, which correlated positively with elevated production of granzyme B (**Supplementary Fig. S24**). Again, SAProsome-3 was proven to be safe for intravenous administration as supported by stable body weights in animals (**Fig. 6d**).

Fig. 6. SAProsome-3 potentiates therapeutic efficacy and synergizes with checkpoint blockade inhibition. a–d LLC tumor-bearing mice were treated with free MSA-2 via intravenous (IV) administration or oral gavage, and SAProsome-3 following intravenous injection at indicated MSA-2–equivalent doses. Tumor growth curves a, spider plots of

individual tumor growth curves **b**, Kaplan–Meier survival curves **c** and body weight changes **d** of tumor-bearing mice are shown ($n = 6$ mice/group). **e–h** Mice with $\sim 150 \text{ mm}^3$ subcutaneous B16F10 tumors were administered with a single dose of saline (IV), free MSA-2 (PO) or SAProsome3 (IV) equivalent to 35 mg/kg MSA-2. For combined treatment with the checkpoint blockade antibody, mice were additionally administered with anti-PD-L1 antibody intraperitoneally three times at a dose of 5 mg/kg ($n = 6$ mice/group). **f** Tumor-growth curves for individual mice. **g** Kaplan–Meier survival analysis (log-rank test). The mice were treated with the indicated formulations, and a tumor volume of 2000 mm^3 was set as the endpoint criteria. **h** Body weight change of B16F10 tumor-bearing mice during treatment ($n = 6$ mice/group).

2. Another important control is MSA-2 loaded into liposomes; i.e., not a prodrug form. The authors claim that they were not able to load MSA-2 into liposomes and it is logical that converting into the prodrug format would increase lipophilicity and loading into liposomes. However, the pKa of the carboxylic acid of MSA-2 is 4.8 and therefore it will also be water insoluble at lower pH values and so presumably would load well into liposomes under these conditions. Have the authors attempted such alternatives for loading of MSA-2 into liposomes so that a comparison between a liposomal formulation and a prosome can be made? Such a study is important for determining the relative importance of the prodrug feature relative to the liposomal delivery system.

Response:

As the reviewer notes, including unmodified MSA-2 formulated in liposomes is an important control. In response to the reviewer's suggestion, we attempted to encapsulate free MSA-2 into liposomes under acidic conditions by adjusting the pH of the media, and the results are presented in the below Figure R1. We observed visible precipitates when free MSA-2 molecule at different pH values was attempted to be formulated into liposomes. In contrast, the solution containing SAProsome-3 at a concentration equivalent to 2 mg/mL MSA-2 remained stable before and after centrifugation. These results verified that unmodified MSA-2 is not miscible with liposomal compositions and the prodrug strategy is necessitated to alter the physicochemical property of MSA-2 for effective liposomal delivery.

Figure R1. A comparison was made between the water dispersion of MSA-2 and prodrug **3** when fabricated into liposomes. It was observed that at a concentration of 1 mg/ml and different pH values, MSA-2 liposomes produced precipitates in the solution, both before and after centrifugation (5000 rpm, 10 min). In contrast, the liposomal formulation of prodrug **3** (SAProsome-3) remained transparent even at a higher concentration of 2 mg/ml and pH7.4. S1: sample 1, S2: sample 2, S3: sample 3.

3. The authors claim that uncontrolled systemic inflammation is an issue for systemically administered STING agonists and that their design might mitigate this problem. However, the authors do not quantify levels of serum cytokines or the levels of STING-driven inflammation in major distribution organs and so the relative impact of the design on systemic inflammation compared to delivery of a free MSA-2 agonist is not known. These experiments need to be performed to help determine if the design can minimize systemic inflammatory effects.

Response:

According to this reviewer's comment, we quantified the levels of serum cytokines and STING-driven inflammation in major distribution organs to determine the potential of our design in minimizing systemic inflammatory effects compared to free MSA-2 agonist delivery. We confirmed the doses of oral MSA-2 at 160 mg/kg, intravenous MSA-2 at 40 mg/kg, and intravenous SAProsome-3 at 30 mg/kg (MSA-2 equivalence), which had comparable antitumor efficacy in the LLC model (**Fig. 6a-c**). Hence, a 1.5-fold higher dose than therapeutic dose was injected for evaluation of systemic inflammatory effects. In addition, toxicities to major organs using serum biochemical analysis and histological examination were also

assessed. The results and corresponding descriptions have been added to the revised manuscript (Page 14-15).

Nanoliposomal vesicles abrogate STING agonist-associated systemic toxicities

Finally, we investigated the potential toxicity induced by systemic administration of STING agonists. The same administration schedule with a 1.5-fold higher dose than therapeutically relevant doses used in the LLC model was tested. Serum biochemical analysis showed that following SAProsome-3 treatment, all parameters remained unchanged as compared to the healthy group (**Fig. 7f-g**). Conversely, the animals receiving intravenous MSA-2 experienced significant increases in aspartate aminotransferase (AST) and alanine aminotransferase (ALT), indicating obvious liver toxicities. Moreover, oral administration of MSA-2 also resulted in significantly elevated serum enzyme levels, including AST, ALT, urine acid (UA) and blood urea nitrogen (BUN). Histological examination of organ sections also suggested obvious toxicities to liver and kidney tissues after treatment with free MSA-2 *via* either intravenous or oral administration, while those treated with SAProsome-3 were indistinguishable from untreated animals (**Supplementary Fig. S26**). We found acute inflammation in free drug-treated groups, characterized by elevated levels of inflammatory cytokines/chemokines (**Fig. 7h** and **Supplementary Fig. S27**). Of note, IL-6 (a key driver of cytokine storm) maintained a 12.5 or 18.6-fold higher level in serum from mice treated with MSA-2 orally or intravenously compared to that of SAProsome-3 (**Fig. 7h**). In contrast, SAProsome-3 showed negligible fluctuation of these cytokines/chemokines in serum and healthy organs, indicating minimized systemic inflammation and immunotoxicities.

Figure 7. f-g C57BL/6 mice were treated with free MSA-2 orally at a dose of 240 mg/kg or intravenously at a dose of 60 mg/kg, and SAProsome-3 intravenously at a dose of 45 mg/kg per mouse, every three days for a total of three injections. Blood chemistry of mice were analyzed on day 8 postadministration. **h** The levels of IL-6 in the serum and other tissues were determined using ELISA at 6 h postadministration of free MSA-2 (PO, 240 mg/kg), free MSA-2 (IV, 60 mg/kg) or SAProsome-3 (IV, 45 mg/kg) (n = 5).

Meanwhile, the quantitative levels of TNF- α and IFN- γ in the serum and other tissues have been presented in the supplementary Figure S27.

Fig. S27. a-b The levels of TNF- α and IFN- γ in the serum and other tissues were determined using ELISA 6 h after a single administration of free MSA-2 (PO, 240 mg/kg), free MSA-2 (IV, 60 mg/kg) or SAProsome-3 (IV, 45 mg/kg) (n = 5 in each group).

4. Related to above, while the weight loss data provided suggests that the system is well-tolerated, additional analysis of toxicity, including serum markers of liver/kidney damage as well as organ pathology should be performed to clearly demonstrate relative toxicity of the system. Also, the body weight data (Fig 3C) should instead be plotted as % loss or gain from original mass.

Response:

Related to above response, we have assessed serum markers of liver/kidney damage, as well as organ pathology. While free MSA-2 demonstrated effective antitumor efficacy, a

higher dose was required than the prodrug liposome to elicit comparable activity, which may cause obvious dose-limiting side effects. In contrast, SAProsome-3 showed negligible *in vivo* toxicities such as liver and renal damages even when administered at 45 mg/kg (MSA-2 equivalence). Thus, the prosome formulation was expected to optimize the *in vivo* delivery of MSA-2 agonist against cancer, triggering substantial antitumor immune responses while without inducing systemic toxicities. This prosome platform could be beneficial for further clinical translation. We have included these results and relevant descriptions in the revised manuscript (Page 14-15).

Nanoliposomal vesicles abrogate STING agonist-associated systemic toxicities

Finally, we investigated the potential toxicity induced by systemic administration of STING agonists. The same administration schedule with a 1.5-fold higher dose than therapeutically relevant doses used in the LLC model was tested. Serum biochemical analysis showed that following SAProsome-3 treatment, all parameters remained unchanged as compared to the healthy group (**Fig. 7f-g**). Conversely, the animals receiving intravenous MSA-2 experienced significant increases in aspartate aminotransferase (AST) and alanine aminotransferase (ALT), indicating obvious liver toxicities. Moreover, oral administration of MSA-2 also resulted in significantly elevated serum enzyme levels, including AST, ALT, urine acid (UA) and blood urea nitrogen (BUN). Histological examination of organ sections also suggested obvious toxicities to liver and kidney tissues after treatment with free MSA-2 *via* either intravenous or oral administration, while those treated with SAProsome-3 were indistinguishable from untreated animals (**Supplementary Fig. S26**). We found acute inflammation in free drug-treated groups, characterized by elevated levels of inflammatory cytokines/chemokines (**Fig. 7h** and **Supplementary Fig. S27**). Of note, IL-6 (a key driver of cytokine storm) maintained a 12.5 or 18.6-fold higher level in serum from mice treated with MSA-2 orally or intravenously compared to that of SAProsome-3 (**Fig. 7h**). In contrast, SAProsome-3 showed negligible fluctuation of these cytokines/chemokines in serum and healthy organs, indicating minimized systemic inflammation and immunotoxicities.

Figure 7. f-g C57BL/6 mice were treated with free MSA-2 orally at a dose of 240 mg/kg or intravenously at a dose of 60 mg/kg, and SAProsome-3 intravenously at a dose of 45 mg/kg per mouse, every three days for a total of three injections. Blood chemistry of mice were analyzed on day 8 postadministration. h The levels of IL-6 in the serum and other tissues were determined using ELISA at 6 h postadministration of free MSA-2 (PO, 240 mg/kg), free MSA-2 (IV, 60 mg/kg) or SAProsome-3 (IV, 45 mg/kg) (n = 5).

Fig. S26. Representative images of H&E staining of major organs (heart, lung, liver, and kidney) excised from C57BL/6 mice after different treatments.

Additionally, we have represented the change in body weight more accurately by plotting the body weight data as % loss in the revised manuscript (**Figure 3c**).

5. Considering that the rest of the studies, including biodistribution studies, were performed in mice, it was not clear why the pharmacokinetic analysis was done in rats. It would have been much more instructive to do this in mice so that relationships between pharmacological properties, efficacy, and toxicity can be drawn.

Response:

Many thanks for your suggestions. Using rats for the pharmacokinetic analysis was based on the requirement of sufficient blood samples for subsequent HPLC analysis. Collecting approximately 500 μ L of blood sample in mice needs to sacrifice one mouse at each time point, which is unfavorable for ethical and animal welfare reasons. Therefore, to reduce the number of experimental animals, we have previously used rats that have sufficient blood volume for analysis to measure drug plasma concentrations (Please refer to our work: Proc Natl Acad Sci U S A 120, e2210385120 (2023); Adv Sci, e2204890 (2023); Biomaterials 290, 121814 (2022); J Control Release 328, 237-250 (2020); ACS Nano 16, 10242-10259 (2022); Am J Transplant 21, 3871-3882 (2021); etc.). In addition, there are examples that used rats for pharmacokinetic analysis (e.g., Nature Communications, 2023, 14.1: 255; Nature communications, 2021, 12.1: 3453; Nature Biomedical Engineering, 2019, 3.4: 264-280; Nature biomedical engineering, 2021, 5.3: 252-263; Science advances, 2020, 6.45: eabc1725; ACS nano, 2022, 16.12: 21417-21430; Proc Natl Acad Sci U S A 2011, 108.5: 1850-1855; etc.).

We also agree with the reviewer that further experiments and evaluations are necessary

to comprehensively understand the relationships between pharmacological properties, efficacy, and toxicity. The suggestion to use the same species for both pharmacokinetic and biodistribution studies is highly appreciated.

6. Related to above, the authors also need to evaluate the biodistribution of drug in other distribution organs such as liver, spleen, and potentially lung; only tumor and LNs are shown and, while these are important, the ratio of drug in these tissues relative to that in other organs is more important. Further, in the original Pan et al. paper it is shown that free MSA-2 preferentially accumulates in tumor tissue due to change in solubility due to protonation of the carboxy group within the tumor microenvironment. So, an evaluation of drug distribution to other organs and the ratio of drug in tumor/other tissues needs to be performed.

Response:

Thanks for this valuable suggestion. We investigated the biodistribution of the drug in other organs, including the heart, liver, spleen, lung, and kidney in the MC38 model at 8 hours post-administration. Our results demonstrated that SAProsome-3 exhibited a higher accumulation in the tumor and LNs, while showing lower distribution in the heart and kidney, in comparison to free MSA-2. Remarkably, in the tumor of mice treated with SAProsome-3, MSA-2 and its intact form maintained a 75.3 and 73.8-fold higher concentration compared to that in the heart and kidney, respectively (**Fig. 7c** and **Supplementary Fig. S25**). In contrast, the concentration of MSA-2 in the tumor was lower than the concentration in the heart and kidney of mice treated intravenously with free MSA-2.

Also, there are no quantitative data of drug concentration in other tissues except for the tumor and plasma in the original Pan et al. paper. We found that the MSA-2 concentration in the tumor at 4 h after oral administration at a dose of 60 mg/kg was approximately 0.2 μM (i.e., 59 ng/g), which was comparable to our data of the MSA-2 concentration (i.e., 10 ng/g) at a dose of 17.5 mg/kg at 8 h after oral administration, indicating the reproducibility and reliability between these data.

In addition, we have calculated the ratios of drug concentrations in tumors *versus* other organs for SAProsome-3 and free MSA-2 (administered either intravenously or orally). The data revealed significant increases in the tumor/other tissues ratios in the SAProsome-3 group compared to both free MSA-2 groups, manifesting the preferential accumulation of SAProsome-3 in target tumor lesions and unfavorable tissue biodistribution of free drugs.

Therefore, these data suggest that the prodrug-formulated liposome (i.e., SAProsome-3) has a favorable distribution profile that can potentially increase intratumoral drug accumulation while minimizing drug exposure in off-target organs. We have included these results and analysis in the revised manuscript on Page 14, as follows.

On the other hand, SAProsome-3 had low distribution across other normal organs, whereas intravenously administered MSA-2 accumulated at a higher concentration in the heart and kidneys (**Supplementary Fig. S25**).

Figure S25. Mice bearing MC38 xenograft tumors were given a single dose of MSA-2 or SAProsome-3 at a 17.5 mg/kg of MSA-2 equivalent dose. Drug concentration in major organs was determined using HPLC analysis at 8 h after administration. Ratios of drug concentrations in tumors *versus* other tissues (e.g., heart, liver, spleen, lung and kidney) are shown in purple on the bars. Data represent the means \pm SD (n = 6).

7. In Figure 2C, the activity of prosomes vs. free MSA2 are shown at 2 different concentrations. First, this (or similar) should be done over a wider range of concentrations and an EC50 estimated for the different groups; this will allow for the relative potencies of each to be determined, which is an important metric when comparing new formulations. Second, statistical analysis of this data is missing.

Response:

We appreciate the reviewer’s constructive comments and have conducted additional experiments to determine the EC₅₀ values and relative potency of SAProsomes and free MSA-2. We used various concentrations of MSA-2 and SAProsomes to measure IFN- β secretion in

cell culture and calculated the EC₅₀ values using the GraphPad Prism software.

We also conducted statistical analysis of the data in **Fig. 2c** using a one-way ANOVA analysis, and the results showed significant differences between SAProsomes and free MSA-2 at both tested concentrations in terms of *IFNβ1* mRNA expression. These new data were incorporated into **Fig. 2d** and relevant description was presented in the revised manuscript (Page 6) as follows.

In addition, dose-dependent secretion of the critical cytokine, IFN-β, after treatment with SAProsomes was observed in BMDCs (Fig. 2d), which allowed to identify the superior activity of SAProsome-3 and 4 with relatively low half maximal effective concentration (EC₅₀).

Figure 2. c STING-dependent downstream expression of inflammatory cytokines in BMDCs treated with MSA-2 or SAProsomes as assessed via qRT-PCR. **d** Dose–response curves of IFN-β secretion upon various treatments. The concentration of IFN-β in BMDC culture medium was determined via ELISA.

8. In Figure 2I, a control should be run using either non-Ova-pulsed DCs and/or B16 cells that do not express Ova in order to demonstrate that the killing response is antigen specific. It is somewhat surprising that there would be a sufficient number of Ova-specific T cells in a wild-type mouse that can be expanded in vitro to kill target cells. Therefore, it is possible that what is being observed is non-specific target cell killing; the controls mentioned above are necessary to demonstrate this.

Response:

We agree with the reviewer’s comment to include additional controls to validate the antigen-specific killing response presented in Figure 2I. To address this concern, we included B16F10 cells that do not express OVA as a control. Our results demonstrated a significant decrease in the killing effect on these control cells relative to the treatment of B16F10-OVA,

suggesting that the enhanced killing response is mediated *via* the OVA antigen. We have supplemented these results along with the relevant discussion in the revised manuscript (Page 7) as follows.

Crucially, pretreatment of BMDCs with SAProsomes and a tumor-specific antigen (e.g., OVA) significantly enhanced the cytotoxicity of splenic T-cells against tumor cells (B16F10-OVA), as compared to untreated cells (**Fig. 2h**). In contrast, splenic T-cells had limited cytotoxicity against B16F10 tumor cells. These experimental results indicate that intracellular esterase can spontaneously trigger the liberation of active MSA-2, which eventually facilitate the activation of STING pathway and lead to an augmented expression of cytokines.

Figure 2. h Cotreatment with the OVA peptide and SAProsomes increased the B16F10-OVA cell lysis via cytotoxic T lymphocytes *in vitro*. Experimental protocol is shown on left-hand side. Cell cytotoxicity was determined based on the lactate dehydrogenase (LDH) concentration in medium supernatant, as measured by the LDH assay. The cell number ratios (T-cell/B16F10 or B16F10-OVA cell/BMDC) were fixed at 10:1:5. Statistical significance was analyzed using one-way analysis of variance.

9. A technical consideration is why the percentage of CD86+CD80+ DCs is so high (~25%) in the untreated group (Figure 2G); this may suggest a need to better titrate antibodies or perhaps another source of innate immune activation (e.g., cell stress, contaminants in cell culture).

Response:

We have carefully examined our experimental protocols and found no indication of contamination or cell stress. However, to address the concern raised by the reviewer, we optimized our incubation time of BMDCs from 8 days to 6 days to reduce the innate immune activation. We performed additional experiments to revalidate the percentage of CD86+CD80+ DCs, and the results showed decreased levels of CD86+CD80+ DCs in the

untreated group. We appreciate the reviewer's comment, and we have updated the results in the revised manuscript.

Figure 2. f Following treatment with SAProsomes, BMDCs exhibit significantly increased activation markers, including major histocompatibility complex II (MHC II) **e** and CD80/CD86 **f** (n = 3).

10. A challenge with ester-based prodrugs is potential to be cleaved by murine esterases but not human esterases. The authors should confirm prodrug activity in human cells as well.

Response:

Thanks for the comment. As the reviewer notes, validation of ester-based prodrugs by human esterases is necessary. We therefore conducted experiments in the human monocytic cell line THP1 to assess the cleavability and activity of prodrug **3**. The following experiments were used to compare the STING-activating activity of free MSA-2 and SAProsome-3. The *in vitro* results indicate that prodrug **3** can be spontaneously cleaved to release active MSA-2 in response to human esterase and to trigger the activation of STING pathway. We have incorporated these findings and descriptions in the revised manuscript (Page 6-7).

A human monocytic cell line THP1 was further utilized to validate the potency of SAProsomes in activating STING. The results found that SAProsome-3 exhibited the fast intracellular breakdown of prodrug **3** and release of MSA-2 as shown in **Supplementary Fig. S12**. Moreover, SAProsome-3 was able to upregulate the mRNA expression of IFN- β , TNF α and CXCL10 that are regulated by STING, as well as activate proteins associated with the STING pathway in THP1. Notably, SAProsomes also significantly enhanced IFN- β secretion, which indicated their potential as the potent STING activator (**Supplementary Fig. S13**).

Fig. S12. **a** HPLC analysis was used to measure drug activation in THP1 cells. The cells were incubated with SAProsome-3 for either 1 or 2 hours. **b** Percentages of activated MSA-2 in THP1 cells were determined using their corresponding standard curves. The data are presented as the mean \pm SD ($n = 3$).

Fig. S13. **a–b** Human THP1 cells were incubated with either free MSA-2 or SAProsome-3 for 6 hours. The activation of the STING pathway was further analyzed using western blot analysis **a** and quantitative real-time PCR analysis **b** of gene expression. **c** IFN- β levels in cell culture media of human THP1 treated with the indicated concentrations of MSA-2 formulations were determined by ELISA.

11. A more minor point is that the authors are encouraged to expand upon their discussion as to why prosome3 seems to have the best therapeutic efficacy of the 4 tested. This is an interesting finding and so the authors should offer some reasons as to why this might be the case.

Response:

We sincerely thank the reviewer for this comment. We have included relevant discussions in the revised manuscript (Page 17-18) as follows.

We disclosed that the overall chemical structures of prodrugs affected the hydrolysis kinetics and activation of MSA-2 in response to esterase. In fact, a positive correlation between hydrolysis rate and STING-stimulating activity was observed in this study. Moreover, SAProsome-3 had the highest AUC_{0-t} , which could, in part, explain its superior antitumor efficacy. Despite these findings, the structure-activity relationships remain to be explored in depth.

REVIEWERS' COMMENTS

Reviewer #1 (Remarks to the Author):

The authors have done an admirable job responding to the review critiques with additional data and discussion. I feel the paper is substantially approved and recommend acceptance.

Response: We appreciate the positive comments of the reviewer.

Reviewer #2 (Remarks to the Author):

The authors have adequately addressed all of my comments and have included a number of new experiments that have strengthened the conclusions and rigor of the work. Overall, this work appears to represent an important advance in enhancing the efficacy of systemically administered STING agonists.

Response: We appreciate the positive comments of the reviewer.